# Mechanism of Anticancer Action of Novel Imidazole Platinum(II) Complex Conjugated with G2 PAMAM-OH Dendrimer in Breast Cancer Cells

**DOI:** 10.3390/ijms22115581

**Published:** 2021-05-25

**Authors:** Robert Czarnomysy, Anna Muszyńska, Jakub Rok, Zuzanna Rzepka, Krzysztof Bielawski

**Affiliations:** 1Department of Synthesis and Technology of Drugs, Medical University of Bialystok, Kilinskiego 1, 15-089 Bialystok, Poland; anna.muszynska@umb.edu.pl (A.M.); kbiel@umb.edu.pl (K.B.); 2Department of Pharmaceutical Chemistry, Faculty of Pharmaceutical Sciences in Sosnowiec, Medical University of Silesia, 41-200 Sosnowiec, Poland; jrok@sum.edu.pl (J.R.); zrzepka@sum.edu.pl (Z.R.)

**Keywords:** breast cancer, PAMAM, dendrimer, platinum(II)

## Abstract

Transition metal coordination compounds play an important role in the treatment of neoplastic diseases. However, due to their low selectivity and bioavailability, as well as the frequently occurring phenomenon of drug resistance, new chemical compounds that could overcome these phenomena are still being sought. The solution seems to be the synthesis of new metal complexes conjugated with drug carriers, e.g., dendrimers. Numerous literature data have shown that dendrimers improve the bioavailability of the obtained metal complexes, solving the problem of their poor solubility and stability in an aqueous environment and also breaking down inborn and acquired drug resistance. Therefore, the aim of this study was to synthesize a novel imidazole platinum(II) complex conjugated with and without the second-generation PAMAM dendrimer (PtMet2–PAMAM and PtMet2, respectively) and to evaluate its antitumor activity. Cell viability studies indicated that PtMet2–PAMAM exhibited higher cytotoxic activity than PtMet2 in MCF-7 and MDA-MB-231 breast cancer cells at relatively low concentrations. Moreover, our results indicated that PtMet2–PAMAM exerted antiproliferative effects in a zebrafish embryo model. Treatment with PtMet2–PAMAM substantially increased apoptosis in a dose-dependent manner via caspase-9 (intrinsic pathway) and caspase-8 (extrinsic pathway) activation along with pro-apoptotic protein expression modulation. Additionally, we showed that apoptosis can be induced by activating POX, which induces ROS production. Furthermore, our results also clearly showed that the tested compounds trigger autophagy through p38 pathway activation and increase Beclin-1, LC3, AMPK, and mTOR inhibition. The high pro-apoptotic activity and the ability to activate autophagy by the imidazole platinum(II) complex conjugated with a dendrimer may be due to its demonstrated ability to reverse multidrug resistance (MDR) and thereby increase cellular accumulation in breast cancer cells.

## 1. Introduction

The ever-increasing incidence of cancer seems to be a disturbing and worldwide health problem. From all cancers in women, breast cancer (BC) currently ranks first in the number of new cases, and this amount dangerously increases every year. In 2020, there were an estimated 2,261,419 new cases of this neoplasm and 684,996 deaths, accounting for 11.7% and 6.9% of all cancers, respectively [1]. The decrease in mortality associated with the above disturbing data can be achieved by improving early diagnosis or developing novel methods for the precise treatment of various molecular subtypes of cancers. Therefore, there is an urgent need to introduce novel and more active therapeutic strategies.

Chemotherapy is one of the therapeutic approaches used to treat cancer. The drug used for the systemic treatment of BC, especially in its triple negative variant is cisplatin (cisPt), representing the platinum complexes class [2,3]. Unfortunately, this drug has many severe side effects, such as kidney, liver, heart, bone marrow, and nervous system toxicity. Moreover, cisPt exhibits the highest emetogenic potential among all cytostatic drugs that significantly limits its effective therapeutic dose and use in clinical treatment [4]. However, a much more important problem associated with cytostatic treatment is the phenomenon of multidrug resistance (MDR). It turns out that despite a good initial response to cisPt application in BC patients, resistance to this drug develops over time and thus negatively affects its therapeutic efficacy [4,5]. Resistance to cisPt develops at multiple levels of the neoplastic cell and its environment. Several mechanisms are involved in this process that affect the amount of drug in the cancer cell (accumulation and metabolism), the apoptotic process, DNA damage repair, or the tumor microenvironment [2,4,6,7]. In the case of intracellular drug accumulation, there are two main mechanisms of resistance, which are related to a decrease in the influx of the drug into the cell or its increased efflux [2,7]. Decreased drug uptake is a result of, among other things, a downregulation of copper transporter 1 (CTR1) and/or an increase in copper transporter 2 (CTR2) expression, which is responsible for the degradation of CTR1 [2]. In turn, the intensified elimination of cisPt from the cell is due to elevated levels of copper-transporting ATPase 1 and 2 (ATP7A and B), as well as the up-regulation of ATP-binding cassette (ABC) transporters, i.e., multidrug resistance protein 1 (MDR1, P glycoprotein), multidrug resistance-associated proteins (MRPs), or breast cancer resistance protein (BCRP) [2,8]. Furthermore, the overexpression and excess of the cisPt detoxifying enzymes (glutathione (GSH) and metallothioneins (MTs)) are also responsible for the depleted drug concentrations in the cell [2,3,6,7]. From this point of view, it is important to obtain drugs that would have less toxicity to normal cells than cisPt and will overcome multidrug resistance.

An interesting alternative to the commonly used platinum(II) complexes in treatment seems to be berenil derivatives of this transition metal. In our study, these compounds exhibited greater cytotoxicity than cisPt against human breast cancer cell lines MCF-7 and MDA-MB-231. The apoptosis induced by these compounds appears to occur via the intrinsic mitochondrial pathway as evidenced by a decrease in mitochondrial membrane potential (MMP). Moreover, these derivatives caused an increase in intracellular topoisomerase IIα levels, which was a reason for the increased DNA fragmentation and directing the cell to the apoptotic pathway. It is also worth noting that in an in vivo study using MCF-7 and MDA-MB-231 xenografts in zebrafish, these compounds significantly inhibited tumor growth, demonstrating their anticancer activity [9]. Unfortunately, the synthesized berenil complexes of platinum(II) were characterized by high hydrophobicity, which may significantly limit the bioavailability of these substances in vivo. For this reason, it would be appropriate to search for methods that could increase their solubility in water and thus enhance their bioavailability.

The problem with the application of many drugs is their low bioavailability, which is particularly linked to poor solubility or permeability through the biological membranes of the used agent. In the context of anticancer drugs, it is also very important to minimize the side effects that are a result of toxic activity on normal body cells and to reduce the dose of the applied drug so that these symptoms do not occur. Nanomedicine and drug delivery systems, such as dendrimers, seem to be very promising in overcoming these difficulties [10,11,12]. Dendrimers are polymeric molecules that are characterized by a three-dimensional, branched structure similar in shape to a sphere with a size of 2 to about 20 nm [13]. In their structure, we can distinguish a multifunctional core from which numerous branches are diverging radially and end with free functional groups enabling modification of the molecule or conjugation of other compounds. There are two ways of drug interaction with this nanocarrier: by enclosing (encapsulation) the substance in cavities located inside the dendrimer structure or by the formation of an ester or amide bond between the drug molecule and the terminal free functional groups [10,13]. The first synthesized and currently the most widely used is the polyamidoamine (PAMAM) dendrimer [10]. An equally important aspect of these macromolecules is their cytotoxicity. It turns out that this is determined by the generation of the dendrimer (if the generation is higher, the structure becomes more toxic) and the nature of its surface (positive charge resulting from the occurrence of amino groups destabilizes cell membranes). In this context, structures with a neutral or negative charge are less toxic [11,13]. Interestingly, these structures can be cytotoxic by themselves against cancer cells, and their mechanism of action is the induction of apoptosis resulting from the formation of ROS and DNA damage [11]. Moreover, there is evidence that the use of these nanocarriers loaded/conjugated with an anticancer drug can overcome MDR [12].

Dendrimers appear to exhibit a multitude of properties that may have positive effects on the bioavailability of a substance while simultaneously reducing its dose and consequently the adverse effects caused by it. A very important property of these nanoparticles is their ability to overcome MDR. Therefore, the aim of our study was to synthesize a novel imidazole platinum(II) complex conjugated with the second-generation PAMAM dendrimer and evaluate the effect of these compounds on anticancer activity against human breast cancer cell lines MCF-7 and MDA-MB-231.

## 2. Results

### 2.1. PtMet2 and PtMet2–PAMAM Inhibit Growth and Viability of Breast Cancer Cells

The compounds synthesized in this study, PtMet2 and PtMet2–PAMAM (Figure 1), were screened for their in vitro cytotoxic activities against breast cancer cell lines (MCF-7 and MDA-MB-231) and normal human breast epithelial cells (MCF-10A) using MTT assay upon 24 h of exposure [9]. Cisplatin was used as a reference compound. Both synthesized compounds were characterized by high cytotoxic activity against two breast cancer cell lines, with PtMet2–PAMAM being the more active compound (Figure 2). An important note is the fact that in the case of normal breast cells, the tested compounds did not show higher cytotoxicity than in the case of breast cancer cells. The IC_50_ values for PtMet2 and PtMet2–PAMAM compounds were >5.0 μM and 0.86 μM in MCF-7 cells, while they were >5.0 μM and 0.48 μM in MDA-MB-231 cells, respectively. In the case of cisplatin in MCF-7, the IC50 was >5.0 μM; in MDA-MB-231, it was >5.0 μM. In MCF-10A cells, the obtained IC_50_ values were >5.0 μM for PtMet2, 0.99 μM for PtMet2–PAMAM, and >5.0 μM for cisplatin. It should be noted that although the IC_50_ values for PtMet2 and cisplatin look similar, their cytotoxic activity is quite different, as indicated by the graphs in Figure 2. The PtMet2 compound is definitely more active than cisplatin.

### 2.2. Antiproliferative Activity of PtMet2 and PtMet2–PAMAM in Zebrafish Model

In order to confirm the high biological activity of the tested compounds, a study of the evaluation of antiproliferative properties was carried out in a zebrafish embryo model (Figure 3). In untreated eggs, we observed normal embryo development, which was revealed in consecutive synchronous cleavages. This process was disturbed in embryos exposed to PtMet2 and PtMet2–PAMAM. At 1.5 hpt (hours post treatment), PtMet2 and PtMet2–PAMAM treated eggs showed a deterioration of division, disorientation, and initial signs of cell fusion. Whereby this phenomenon was more pronounced in the case of PtMet2–PAMAM. Within the next 30 min, fusion dramatically progressed in PtMet2–PAMAM treated eggs as well as PtMet2, while the control eggs continued cell division without any apparent delay. After two hours of incubation with PtMet2–PAMAM, we noticed complete cell fusion and lysis. Cisplatin did not show such strong changes during the study period. No alteration in cell division and development in the untreated control eggs were observed.

### 2.3. PtMet2 and PtMet2–PAMAM Modulate NF-κB Levels

In recent years, studies have established strong support for the critical role of NF-κB in cancer. Abnormally high NF-κB activity is a clinical hallmark of chronic inflammation and has been found in many types of cancer cells. Therefore, drugs that inhibit NF-κB activity have been found to be useful additions to the chemotherapy regimens of a variety of cancers [14]. However, some recent findings have suggested that this generalization should be viewed with caution, because an increase in NF-κB levels may direct cancer cells to cell death [15]. Employing immunofluorescence staining, we investigated the effect of PtMet2 and PtMet2–PAMAM on the levels of expressed NF-κB protein in MCF-7 and MDA-MB-231 cells. As shown in Figure 4, treatment with the tested compounds resulted in an increase in NF-κB protein levels in the cytosolic as well as nuclear fraction. Consistent with these observations, an increase in NF-κB protein levels correlated with an increase in the active antiproliferative tested compounds.

### 2.4. PtMet2 and PtMet2–PAMAM Compounds Induce Apoptosis by Inducing the Extrinsic and Intrinsic Pathway

Apoptosis is an effective mechanism for the induction of cell death in cancer cells [16]. To investigate the effect of the novel imididazole platinum(II) complexes on apoptosis induction (by flow cytometer), both MCF-7 and MDA-MB-231 cells were treated with different concentrations (1.5 μM and 2.5 μM) of PtMet2 and PtMet2–PAMAM for 24 h. To detect apoptosis, the cells treated with the test compounds were dyed with Annexin V-FITC and propidium iodide. PtMet2–PAMAM triggered a significant increase in apoptotic cells compared to PtMet2 (Figure 5). In the case of a concentration of 2.5 μM, this difference is several times, both in MCF-7 and MDA-MB-231 cells. At the same time, it should be noted that despite the lower biological activity of PtMet-2, its pro-apoptotic activity is still more than twice as high as that of cisplatin.

Following the apoptosis assessment, the effect of PtMet2 and PtMet2–PAMAM compounds on apoptosis signaling molecules was examined. We determined the status of apoptotic markers such as Bax, PARP, caspase-3, -8, and -9. Bax is a cytosolic protein which, after moving to the mitochondria and attaching to their outer membrane, reaches an active form and interacts with the proteins of this membrane, creating pores through which cytochrome c flows from the intercellular space, contributing to caspase-9 activation [17]. Moreover, this caspase is also activated by internal stimuli, such as ROS, initiating the intrinsic pathway. In contrast, the caspase initiating the extrinsic pathway is caspase-8, triggered by external stimuli. Caspase-3, on the other hand, is activated in both the intrinsic and extrinsic pathways of apoptosis in response to a signal from activated caspases-8 and -9. It is known as the executive caspase and is responsible for directing the cell toward the apoptotic pathway [18]. Caspase-mediated apoptotic cell death is accomplished through the cleavage of several key proteins required for cellular functioning and survival. PARP-1 is one of several known cellular substrates of caspases. Cleavage of PARP-1 by caspases is considered to be a hallmark of apoptosis [19]. Compared to the control and reference drug (cisplatin), the activity of caspase-3, -8, and -9 was upregulated after 24 h of treatment of MCF-7 and MDA-MB-231 cells with the tested compounds (Figure 6). The compound with the highest activity was PtMet2–PAMAM. We observed a similar relationship in the case of Bax (Figure 7) and PARP (Figure 8). Both novel synthesized compounds activated these proteins to a much greater extent than cisplatin. PtMet2–PAMAM showed a greater ability to induce Bax and PARP than PtMet2. These results correlate with previous studies that showed its higher pro-apoptotic potential.

### 2.5. PtMet2 and PtMet2–PAMAM Induce Oxidative Stress through ROS Production

The generation of endogenous reactive oxygen species was measured by flow cytometer using Intracellular Total ROS Activity Assay. Our data revealed that PtMet2 and PtMet2–PAMAM induce oxidative stress through ROS production in a dose-dependent manner (Figure 9). Both synthesized compounds showed a greater ability to produce reactive oxygen species compared to cisplatin. Moreover, PtMet2–PAMAM induced oxidative stress to a greater extent than PtMet2. In this case, the number of ROS cells positive for MCF-7 cells was about 27% (1.5 μM) and 40% (2.5 μM), and for MDA-MB231, it was 27% (1.5 μM) and 44% (2.5 μM). PtMet2, on the other hand, generated about 9% (1.5 μM) and 16% (2.5 μM) in the case of MCF-7, and 11% (1.5 μM) and 15% (2.5 μM) in the case of MDA-MB-231.

### 2.6. PtMet2 and PtMet2–PAMAM Compounds Induce Autophagy by Activating LC3A/B, Beclin-1, and AMPKβ1/2 and Decreasing mTOR

Over the past few decades, anti-cancer drug development has been based on the induction of apoptosis. However, cancer cells trigger multiple pathways to reverse cancer cells from apoptosis [20]. Therefore, new chemical molecules that would additionally guide the cell on the path of death in other ways are constantly searched for. Activating autophagy along with apoptosis could effectively eliminate this problem. As the posttranslational modifications of LC3A/B and regulation of Beclin-1, AMPKβ1/2 and mTOR correlate with the extent of autophagy [21]; we analyzed these proteins by flow cytometry. Moreover, this analysis was also used to identify the formation of autophagosomes and autolysosomes during autophagy.

After 24 h of incubation, we observed that all the tested compounds induced autophagy in MCF-7 and MDA-MB-231 cells with a different intensity compared to the control (Figure 10). There was 96.5% of non-autophagic cells and 3.5% of autophagic cells in the control population of MCF-7 cells and 96.6% of non-autophagic cells and 3.4% of autophagic cells in the MDA-MB-231 cells. The highest activation of autophagy on both cell lines was exhibited by PtMet2–PAMAM, where we observed 25.2% (1.5 µM) and 50.4% (2.5 µM) of autophagic cells in MCF-7, and 28.0% (1.5 µM) and 51.0% (2.5 µM) in MDA-MB-231 cells. An insignificant intensity of autophagy was observed in cisplatin (2.5 µM: 5.2% of autophagic cells in MCF-7 and 6.0% in MDA-MB-231). In the case of the PtMet2 compound, the activation of autophagy was higher than in cisplatin: 7.8% (1.5 µM) and 10.5% (2.5 µM) of autophagic cells in MCF-7, and 8.9% (1.5 µM) and 11.5% (2.5 µM) in MDA-MB-231, respectively. Additionally, the results revealed that PtMet2 and PtMet2–PAMAM induced processing of post-translational modifications of LC3A/B much more so than the reference cisplatin (Figure 11), which correlates with the results indicating an increased formation of autophagosomes induced by these compounds. The induction of autophagy by the synthesized compounds was further confirmed by the validation of key proteins (AMPKβ1/2, Beclin-1, and mTOR) that this process regulated. Quantitative analysis of flow cytometry demonstrated that cancer cells treated with PtMet2 and PtMet2–PAMAM markedly upregulated AMPKβ1/2 and Beclin-1 (initiation factors for autophagosome formation) as compared to the control and cisplatin (Figure 12 and Figure 13). At the same time, this phenomenon was accompanied by inhibition of the mTOR complex, which is the main regulator of translation and autophagy in the cell (Figure 14). The compound with the highest inhibitory properties was PtMet2–PAMAM. Taken together, these multifactorial anti-cancer effects of PtMet2 and PtMet2–PAMAM appear to inhibit the malignant properties of breast cancer cells through inducing both apoptosis and autophagy.

### 2.7. PtMet2 and PtMet2–PAMAM Induce Autophagy by p38 in Breast Cancer Cells

There is some evidence that p38 is a requirement for cancer cell autophagy initiated by various anticancer agents [22]. Therefore, we investigated the effect of PtMet2 and PtMet2–PAMAM on p38 activity in breast cancer cells by immunofluorescence staining. Figure 15 shows a significant increase in the phosphorylation of p38 in the novel platinum compound-treated breast cancer cells relative to the control, suggesting the involvement of p38 in PtMet2 and PtMet2–PAMAM induced autophagy in breast cancer cells. PtMet2–PAMAM was much more active than PtMet2. In the case of cisplatin, only slight changes in cells with regard to the measured proteins were observed, which proves that the newly synthesized compounds are much more active.

### 2.8. Upregulation of p53 Caused by PtMet2 and PtMet2–PAMAM

As it is known, p53 is a significant suppressor of tumors leading to apoptosis in the most DNA-damaging response agents, mainly concerning the intrinsic pathway [23]. The loss of p53 function is observed in cancer cells, so when these cells are treated with anticancer drugs, apoptosis occurs; therefore, cell death also occurs [24]. Our findings showed that both synthesized compounds increased apoptosis and thereby upregulated p53. Figure 16 clearly shows that PtMet2 and PtMet2–PAMAM caused the activation of the p53 protein and its translocation to the nucleus took place, which may explain the high pro-apoptotic activity of the compounds. This effect is especially visible in the case of PtMet2–PAMAM. In the case of cisplatin, p53 activation is much lower.

### 2.9. PtMet2 and PtMet2–PAMAM Induce Proline Oxidase Expression

Through the generated reactive oxygen species (ROS), proline oxidase (POX) might play an important role in apoptosis, especially in the mitochondrial (intrinsic) pathway [25]. Due to the fact that our previous research showed that PtMet2 and PtMet2–PAMAM generate reactive oxygen species and direct breast cancer cells along the path of apoptosis, we decided to check whether POX is involved in these processes. The results of our research clearly show that POX activation occurred in both MCF-7 and MDA-MB-231 cells under the influence of the newly synthesized compounds (Figure 17). This effect is strongest in the case of PtMet2–PAMAM. In the case of cisplatin, the activation of POX was much lower compared to the imidazole platinum(II) complexes.

### 2.10. PtMet2 and PtMet2–PAMAM Inhibit Drug Efflux Pump in Breast Cancer Cells

To investigate the role of PtMet2 and PtMet2–PAMAM on the chemoresistance of MCF-7 and MDA-MB-231 cells, drug efflux was evaluated using the MDR Assay Kit. In both tested breast cancer cells, Efflux Green Detection Reagent pumped out of the cells was decreased with increasing concentrations of the tested compounds (Figure 18). The lowest level of pumping out of compounds was observed in the case of the use of a BCRP inhibitor (Novobiocin). In this case, the MAF_BCRP_ values were as follows: 14.7 (PtMet2 1.5 μM), 7.2 (PtMet2 2.5 μM) 1.7 (PtMet2–PAMAM 1.5 μM), 0.0 (PtMet2–PAMAM 2.5 μM) for MCF-7; and 5.5 (PtMet2 1.5 μM), 19.9 (PtMet2 2.5 μM), 15.4 (PtMet2–PAMAM 1.5 μM), and 8.7 (PtMet2–PAMAM 2.5 μM) in the case of MDA-MB-231. However, in the presence of an MDR1 inhibitor (Verapamil), these values (MAF_MDR1_) were much higher: 41.0 (PtMet2 1.5 μM), 32.4 (PtMet2 2.5 μM), 19.4 (PtMet2–PAMAM 1.5 μM), 15.1 (PtMet2–PAMAM 2.5 μM) for MCF-7; and 28.4 (PtMet2 1.5 μM), 47.0 (PtMet2 2.5 μM), 40.3 (PtMet2–PAMAM 1.5 μM), and 17.6 (PtMet2–PAMAM 2.5 μM) in the case of MDA-MB-231. In both cases, MAF values were lower in MDA-MB-231 cells than MCF-7, with PtMet2–PAMAM contributing to the greatest inhibition of Efflux Green Detection Reagent. The phenomenon of inhibiting pumping out the synthesized compounds from the tumor cells and their greater accumulation inside may be the reason for their high cytotoxic activity, which was observed in the MTT assy. At the same time, the lower MAF values for the MDA-MB-231 cells are reflected in the lower IC_50_ value compared to MCF-7 cells.

## 3. Discussion

Due to their solubilization properties, dendrimers are in the spotlight as carriers for hydrophobic drugs. Additionally, they possess a number of favorable properties, including a well-defined chemical structure, globular shape, low polydispersity index, biocompatibility, and perfect solubility in a large number of solvents, particularly in water. Moreover, their spherical architecture provides a compact structure (1–10 nm) for transport across biological membranes, which makes dendrimers ideal carriers for enhanced solubility of poorly water-soluble drugs [26]. Therefore, it should come as no surprise that in the course of this work, we attempted to synthesize novel imidazole platinum(II) complexes conjugated with the second-generation PAMAM dendrimer. Our previous research showed that platinum(II) complexes can be an interesting alternative to the currently used platinum drugs [9,27,28,29,30]. However, similar to all metal complexes, especially platinum drugs, they have disadvantages, the most troublesome of which is low bioavailability, which results from their high hydrophobicity [31]. Previous studies have shown that this problem can be solved by using dendrimers as drug carriers [32,33]. In addition, as a result of combining a drug with a dendrimer, an increase in its cytotoxicity is often observed, which allows reducing its dose and, as a result, eliminating many side effects [34].

In our study, we demonstrated that the cytotoxicity of PtMet2 conjugated with PAMAM dendrimer (PtMet2–PAMAM) was clearly higher in the tested concentration range that free PtMet-2. The observed effect was more pronounced for MDA-MB-231 than MCF-7 cells, which was possibly due to selective adhesion of the conjugate to cells lacking ER (estrogen receptor) expression. The IC_50_ values for breast cancer cells demonstrated that the PtMet2–PAMAM conjugate (0.86 µM for MCF-7 and 0.48 µM for MDA-MB-231) showed a rise in selectivity and therapeutic effect when compared not only to cisplatin (>5 µM for MCF-7 and MDA-MB-231) but also to the free PtMet2 compound (>5 µM for MCF-7 and MDA-MB-231). This indicates that there is a possibility of a dose reduction while maintaining the therapeutic effect and selectivity, which can protect from the side effects caused by platinum drugs. On the other hand, when comparing MCF-10A cells to MCF-7 cells, the selectivity is negligible. In this case, for both cell lines, the IC_50_ values are at a very similar level (0.99 µM for MCF-10A and 0.86 µM for MCF-7). This may indicate that while the dendrimer conjugate improves delivery, it does not improve selectivity in MCF-10A and MCF-7 cells. We observed a different situation in the case of MDA-MB-231 cells. In this case, the difference between the IC_50_ values is greater (0.99 µM for MCF-10A and 0.48 µM for MDA-MB-231).

Recently, an increased effectiveness of anticancer drugs (cisplatin and doxorubicin) conjugated with dendrimer was also observed by Xue-Ling et al. [35]. The cytotoxic effect is inextricably linked with antiproliferative activity. Therefore, the antiproliferative activity of the novel imidazole platinum(II) complexes with and without a dendrimer derivative was also confirmed in a zebrafish embryo model. Genetic similarities with humans lead to zebrafish often being used as a strong and cost-effective animal model for cancer research and the discovery of cancer drugs. The small size, transparency of zebrafish embryos, easy manipulation, short testing period, and small amount of tested drugs are undoubtedly important aspects giving zebrafish an advantage over the mouse/rat model of cancer [36]. An antiproliferative test showed that PtMet2–PAMAM possessed the most biological activity, which was much higher than that of PtMet2. This compound blocked cell division in a zebrafish embryo model at the four-cell stage of embryonic development, and these disorders were noticeable earlier than in the case of the agent without a dendrimer. Two hours of incubation led to complete cell fusion and lysis in groups exposed to an imidazole platinum(II) complex functionalized dendrimer. The cell division blockade and developmental arrest caused by this compound were irreversible. The results of this experiment are consistent with the results obtained in the cytotoxicity test, which clearly indicates that the dendrimer-conjugated complex shows much higher biological activity at much lower concentrations.

Treatment of cancer cells with chemotherapeutic drugs and the related phenomenon of antiproliferation often results in substantial heterogeneity in the response to NF-κB activity. In some cases, NF-κB activation by chemotherapeutic drugs elicits a pro-survival cellular response and a combination with NF-κB inhibitors improves efficacy [14]. However, some recent reports have challenged this model and proposed that NF-κB activity seen in response to DNA damage induced by ultraviolet radiation and chemotherapeutics can function to promote cell death. The most common stimuli used to reportedly induce pro-apoptotic NF-κB activity are anthracycline, doxorubicin, and its analogues [15]. Ho et al. demonstrated that the treatment of breast cancer cells with doxorubicin generates an NF-κB complex and represses anti-apoptotic gene transcription [37]. Recent reports also showed that Velcade (bortezomib), a drug approved by the Food and Drug Administration for the treatment of multiple myeloma, induces NF-κB, instead of inhibiting as was thought by inducing caspase-dependent mechanisms [38]. This may explain the high pro-apoptotic activity of the tested compounds, especially of the conjugate with the dendrimer. PtMet2–PAMAM turned out to be a strong inducer of both NF-κB and apoptosis. Our results indicated that improving the cell death response to PtMet2 and PtMet2–PAMAM in breast cancer cells depends on stimulating rather than inhibiting NF-κB activity. Our findings suggest that the mechanisms of toxicity of antitumor imidazole platinum(II) complexes in tumor cells may involve interference with NF-κB signaling pathways (especially upregulation), possibly leading to its apoptotic effects.

One way of treating cancer is to gain control or possibly terminate the uncontrolled growth of cancer cells. Using the cell’s own mechanism for death is a highly effective method. Additionally, targeting apoptosis is the most successful non-surgical treatment. Targeting apoptosis is also effective for all types of cancer, as apoptosis evasion is a hallmark of cancer and is nonspecific to the cause or type of cancer. Two common strategies for therapeutic targeting are pro-apoptotic molecule stimulation and anti-apoptotic molecule inhibition [39]. As a result of their modulation, the cell undergoes apoptosis via the extrinsic or intrinsic pathway. In general, tumor cells are more sensitive to the extrinsic pathway than the intrinsic one [40], which indicates that the extrinsic pathway should be targeted for cancer therapy. In the case of the tested compounds, we observed a significant increase in apoptosis induced by both the external and internal pathways. The ability of the compounds to activate caspase-8, demonstrated in this study, provides significant evidence that the compounds induce apoptosis through the extrinsic pathway. Importantly, we also observed an increase in the active form of caspase-9, which is a marker of the intrinsic pathway. The ability of the compounds to activate apoptosis through both pathways is indisputable evidence that one of the molecular targets of the newly synthesized compounds is to activate this type of cell death. At the same time, we noticed that the compound conjugated with the dendrimer showed much greater pro-apoptotic properties. Moreover, the confirmation that PtMet2 and PtMet2–PAMAM have antiproliferative effects just by inducing apoptosis are the results showing these complexes to increase active caspase-3/7 and induce the expression of apoptotic markers BAX and PARP. Our findings are in agreement with previous observations, which showed that platinum(II) complexes induce apoptosis in breast cancer cells through external and internal pathways [9,27,29,30].

Another possible mechanism that could regulate the apoptosis induced by imidazole platinum(II) complexes is oxidative stress. Numerous studies suggest that compounds modulating cellular ROS levels can enhance cancer cell death and sensitize it to certain chemotherapeutic drugs. ROS could act as signaling molecules that trigger oxidative stress, hence leading to enhanced cytotoxicity and apoptosis [41]. In this context, we report here that the induced apoptosis by the synthesized complexes was proportionally associated with a significant increase in ROS production. Previous observations support our current results; these studies showed that cancer cell proliferation inhibition and apoptosis induction are mediated by oxidative stress induction [42]. Importantly, the induction of oxidative stress by metal-based drugs is believed to mediate the activation of autophagy-associated cell death pathway markers [43]. PtMet2 and PtMet2–PAMAM treatment also triggered autophagy, as evidenced by the resulted autophagosomes and the accumulations of LC3A/B and Beclin-1, which are commonly known as methods for monitoring autophagy. LC3A/B and Beclin-1 play indispensable roles in autophagosome formation, and especially, LC3A/B is the most reliable biomarker of autophagy, as it is well established that LC3A/B levels are directly proportional to autophagosome formation [44]. In this study, we demonstrated that imidazole platinum(II) complexes augment LC3A/B and Beclin-1 levels in a dose-dependent manner in MCF-7 and MDA-MB-231 cells, as characterized by flow cytometry. This result is in agreement with recent reports that many anticancer drugs enhanced Beclin-1 and LC3A/B levels in cancer cells [45]. Additionally, the autophagy process is known to be regulated by the MAP kinase signaling pathway, including p38 [46]. In the current study, we found that the PtMet-2 and PtMet2–PAMAM-induced autophagy is associated with activations of the p38 pathway. Moreover, our results also clearly showed that the tested compounds trigger autophagy through AMPK activation and mTOR inhibition. The AMPK increase by imidazole platinum(II) complexes significantly increased autophagy induction, which escalated their cytotoxic effect, demonstrating induced autophagy through AMPK activation and mTOR inhibition. In all the studies on autophagy, it is clear that the conjugate with the dendrimer showed a much greater potential for its induction. We saw similar results with apoptosis. At this stage of the research, we can conclude that the addition of a dendrimer to the compound clearly improves its properties for the induction of cell death, both apoptosis and autophagy. The dual induction of apoptosis and autophagic-associated cell death makes the imidazole platinum(II)-functionalized polyamidoamine dendrimer an ideal therapeutic target for breast cancer.

The p53 protein is a known regulator of both these processes. A proapoptotic function of p53 occurs both at the level of transcription, through the activation of proteins such as PUMA, NOXA, and BAX, and in the cytosol by binding apoptotic proteins such as Bcl-2 and Bcl-xL. Autophagy induction by p53 may either contribute to cell death or constitute a physiological cellular defense response. As with apoptosis, the cellular localization of p53 modulates its impact in autophagy; cytosolic p53 inhibits autophagy while nuclear p53 induces and regulates autophagy [47]. The influence of the tested compounds on the p53 protein clearly showed its translocation to the cellular and nuclear localization in both MCF-7 and MDA-MB-231 cells. We noticed the highest amount of translocated protein in the case of the PtMet2–PAMAM compound, which proves that the compound functionalized with the dendrimer influences the translocation of the p53 protein and therefore apoptosis and autophagy.

Liu et al. showed that during p53-induced apoptosis, there is a significant elevation in POX (proline oxidase), which generates proline-dependent reactive oxygen species (ROS) [48]. POX is a flavin-dependent enzyme associated with the inner mitochondrial membrane. The enzyme catalyzes the conversion of proline into Δ(1)-pyrroline-5-carboxylate (P5C), during which reactive oxygen species (ROS) are produced, inducing intrinsic and extrinsic apoptotic pathways. Due to ROS generation, POX may induce caspase-9 activity, which mediates mitochondrial apoptosis (intrinsic apoptosis pathway). POX can also stimulate TRAIL (tumor necrosis factor-related apoptosis inducing ligand) and DR5 (death receptor 5) expression, resulting in a cleavage of procaspase-8 and thus the extrinsic apoptotic pathway [49]. Liu et al. showed that DLD-1 colorectal cancer cells stably transfected with the POX gene under the control of a tetracycline-inducible promoter and found POX-stimulated expression apoptosis and cleavage of caspase-8 [25]. Similarly, in our studies using breast cancer cells, PtMet2–PAMAM increased the expression of POX in parallel with ROS formation and cell death through the activation of the caspase cascade. All these results reflected that the apoptosis induction by PtMet2–PAMAM may be through the activated POX inducing ROS formation.

In all our studies, the compound conjugated to the dendrimer showed several times higher biological activity compared to the free compound. Reports from recent years show that as a result of the use of PAMAM dendrimers as carriers of commonly used anticancer drugs (methotrexate, doxorubicin, fluorouracil, cisplatin, paclitaxel), the resulting conjugates were characterized by low systemic toxicity, high stability, high water solubility, improved oral bioavailability, increased tumor targeted delivery, increased their retention time, and enhanced antitumor efficacy, which often occurs as a result of a decrease in drug resistance [50].

Multidrug resistance (MDR) in cancer cells is the paramount obstacle for successful chemotherapy. To overcome tumor MDR, higher doses or frequency of dosing of the chemotherapeutic agents are required, thus resulting in the risk of severe adverse side effects or healthy tissue toxicity. MDR is a complex phenomenon that can result from including increased drug efflux by ATP-dependent pumps, such as P-glycoprotein (P-gp; MDR1), breast cancer resistance protein (BCRP), and multidrug resistance-associated protein 1 (MRP1) [51]. Recent studies have shown that nanoparticles (including PAMAM dendrimers) can directly influence drug accumulation in cancer cells. Nanocarriers deliver the chemotherapeutic agent primarily to cancer cells, where it acts to reverse MDR. In contrast to a free drug that can be easily shuttled by the efflux pump out of the cell, nanocarriers can bypass drug efflux transporters on the plasma membrane and therefore support the accumulation of drugs in the cancer cell [50]. Although several mechanisms have been described to explain clinical MDR, up to now, the challenge of therapy resistance treated with anticancer drugs had not been conquered. Thus, there is currently an urgent need to develop effective therapeutic approaches to overcome tumor MDR.

Recently, PAMAM-mediated complexes presented higher advantages on the reversal of tumor MDR. Zhang et al. studied its transportation in resistant breast cancer cells [52]. In their studies, both P-gp and MDR-associated proteins play an important role in the exocytosis process of PAMAM, resulting in its continuous exocytosis in breast cancer cells. In another study, PAMAM-siMDR1 nanocomplexes raised cellular accumulation of doxorubicin and worked synergistically with paclitaxel for treating MDR, which inhibited breast cancer cell growth and induced their apoptosis [53]. Our results are consistent with the above studies, which showed that the dendrimer conjugate inhibits the release of the drug from the cell. PtMet2–PAMAM showed higher cytotoxicity than free PtMet2 in breast cancer cells, which could be induced by more accumulation and a quicker distribution in the nuclei. It could be explained that the diffusion process was slower than endocytosis and the release of PtMet2 from the conjugates was in a sustained manner. Moreover, the conjugate with the dendrimer showed a much greater ability to induce apoptosis and autophagy, which may indicate its ability to interact with biological membranes and enhance the transport of compounds into the cells. Furthermore, the effect of inhibiting drug efflux transporters could help keep the imidazole platinum(II) complex in the cells. In addition, the positive charge of the conjugates benefits entrance into the tumor cells [54]. The final significant increase in the intracellular content of PtMet2–PAMAM could be very helpful for reversing MDR, and the distribution in the nuclei guarantees the tumor-killing effect. The results explain the high cytotoxicity of PtMet2–PAMAM against breast cancer cells. The potential mechanism of antitumor activity of the novel imidazole complex conjugated with a dendrimer is presented in Figure 19.

## 4. Materials and Methods

### 4.1. Materials

Potassium tetrachloroplatinate(II), potassium iodide, acetone, 2-(2-methyl-5-nitro-1H-imidazol-1-yl)acetic acid, PAMAM-OH dendrimer (generation 2), nitric acid(V) silver(I) salt, methanol, diethyl ether, cisplatin, 3-(4,5-dimethylthiazol-2-yl)-2,5-diphenyltetrazolium bromide (MTT), formaldehyde, paraformaldehyde, DMSO, BSA, DAPI, Phalloidin-Atto 565, and Triton X-100 were purchased from Sigma Chemical Co. (St. Louis, MO, USA). Stock cultures of human breast cancer cell (MCF-7 and MDA-MB-231) and normal human breast epithelial cells (MCF-10A) were purchased from the American Type Culture Collection (ATCC, Manassas, VA, USA). Dulbecco’s minimal essential medium (DMEM), fetal bovine serum (FBS), PBS used in a cell culture, trypsin, glutamine, penicillin, and streptomycin were obtained from Gibco (San Diego, CA, USA). An MEGM Mammary Epithelial Cell Growth Medium BulletKit was purchased from Lonza Bioscience (Basel, Switzerland). An FITC Annexin V Apoptosis Detection Kit II, anti-NF-κB mouse monoclonal antibody, anti-p38 mouse monoclonal antibody, anti-PARP mouse monoclonal antibody, anti-POX mouse monoclonal antibody, anti-caspase-3 mouse monoclonal antibody, anti-caspase-8 mouse monoclonal antibody, anti-caspase-9 mouse monoclonal antibody, FITC-labeled secondary anti-mouse antibody, and Stain Buffer were from BD Pharmigen (San Diego, CA, USA). An Autophagy Assay kit, Intracellular Total ROS Activity Assay, and Hoechst 33342 were purchased from ImmunoChemistry Technologies (Bloomington, MN, USA). AMPKβ1/2 rabbit mAb, anti-rabbit IgG (Alexa Fluor 647 Conjugate), mTOR Rabbit mAb (Alexa Fluor 647 Conjugate), and LC3A/B Rabbit mAb (PE Conjugate) were purchased from Cell Signaling Technology (Beverly, MA, USA). An MDR Assay Kit, FITC anti-Bax antibody, anti-Beclin-1 antibody, and Goat anti-mouse IgG (Alexa Fluor^®^ 488) were purchased from Abcam (Cambridge, UK). DAPI was obtained from Thermo Fisher Scientific, Inc. (Waltham, MA, USA). The fluorescence mounting medium was purchased from Agilent Dako (Carpinteria, CA, USA).

### 4.2. Physical Measurements

The structure of the synthesized compound was confirmed by ^1^H NMR and ^13^C NMR spectra recorded on the Bruker AC 200F (Mannheim, Germany) apparatus (^1^H—200 MHz and ^13^C—50 MHz) in deuterated dimethylsulfoxide (d_6_-DMSO). Chemical shifts were expressed as a δ value (ppm). The multiplicity of resonance peaks was indicated as a singlet (*s*), doublet (*d*), triplet (*t*), quartet (*q*), and multiplet (*m*). Infrared spectra were recorded on the Perkin-Elmer Spectrum 100 FT-IR spectrometer (PerSeptive Biosystems, Houston, TX, USA) as KBr pellets (4000–450 cm^−1^). Mass spectra were recorded using a Mariner mass spectrometer (Hiden Analytical, Waltham, MA, USA). Melting points were determined on the Buchi 535 (GER) melting-point apparatus and were uncorrected. Elemental analysis of C, H, and N was performed on a Perkin-Elmer 240 analyzer (PerSeptive Biosystems, Houston, TX, USA), and satisfactory results within ±0.4% of the calculated values were obtained.

### 4.3. Preparation of cis-[Pt_2_(2-(2-methyl-5-nitro-1H-imidazol-1-yl)acetic acid)_4_(berenil)_2_]x4HCl (PtMet2)

To a solution of potassium tetrachloroplatinate (II) K_2_PtCl_4_ (0.72 mmol) in 10 mL of deionized water, 2-(2-methyl-5-nitro-1H-imidazol-1-yl)acetic acid (0.54 mmol) was added, and the mixture was stirred for 48 h. The obtained grayish green residue was filtered off under reduced pressure, washed with deionized water (3 × 3 mL), and dried in a vacuum. To the crude product (dichlorodi(2-(2-methyl-5-nitro-1H-imidazol-1-yl)acetic acid)platinum(II)) (0.227 mmol), silver nitrate (0.454 mmol) dissolved in 3 mL deionized water was added, and the mixture was stirred in the dark at room temperature for 48 h. Then, solid silver chloride was filtered off and washed with deionized water (2 × 1 mL). To the obtained filtrate, berenil (0.227 mmol) and a 10% aqueous sodium chloride solution (3 mL) were added. The mixture was stirred in the dark at room temperature for another two days. The final product was filtered off under reduced pressure and washed with deionized water (2 × 1 mL), 0.1M hydrochloric acid (1 mL), deionized water (2 × 1 mL), and diethyl ether (1 mL) and then dried in a vacuum.

Yield: 36.2% (75.2 mg); yellow powder; mp 258–260^o^C; ^1^H-NMR (DMSO-d_6_) δ (ppm): 12.40 (br, s, COOH), 9.24 (br, s, amidine), 8.03 (s, 4H, Pz), 7.91 (d, *J* = 8.6 Hz, 8H, Ar), 7.68 (d, *J* = 8.3 Hz, 8H, Ar), 4.67 (s, 8H, CH_2_), 3.69 (m, 8H, CH_2_), 2.48 (s, 12H, CH_3_); ^13^C NMR (DMSO-d_6_) δ (ppm): 174.8 (COOH), 165.2 (amidine), 151.80 (Pz), 149.5 (Ar), 138.31 (Pz), 132.84 (Pz), 129.5 (Ar), 122.0 (Ar), 118.0 (Ar), 59.68 (CH_2_), 45.76 (CH_2_), 14.12 (CH_3_); IR (KBr, cm^−1^): 3352 (C=NH imine), 3143 (NH_3_^+^), 3131 (COOH), 2943 (CH_3_), 2858 (CH_2_), 1711 (COOH), 1674 (NCN/C=N imine), 1607 (triazene), 1572 (CN imidazole ring), 1514 (NH_3_^+^), 1510 (N-O), 1485 (CH_2_/CH_3_), 1439 (CH_3_), 1404 (COOH), 1384 (N-O), 1259 (triazene), 854 (Ar-NO_2_), 525 (Pt-N); MS (ES, HR) m/z (M^+^) calcd. for C_52_H_60_Cl_4_N_26_O_16_Pt_2_ 1837.1900, found 1837.1556; Anal. calcd. for C_52_H_56_N_26_O_16_Pt_2_·4HCl·2H_2_O: C, 33.31; H, 3.42; N, 19.43; found: C, 33.36; H, 3.47 N, 19.44.

### 4.4. Preparation of G2 PAMAM-OH dendrimer-(cis-[Pt_2_(2-(2-methyl-5-nitro-1H-imidazol-1-yl)acetic acid(berenil)_2_]) Complex (PtMet2–PAMAM)

PtMet2 (62.1 mg; 0.0338 mmol) and the molar equivalent of PAMAM-OH G2 (80.16 µL, 0.0042 mmol, based on 16 end functional groups) were dissolved in 3 mL of methanol. The mixture was stirred in the dark at room temperature until substrate disappearance (TLC control). After five days, the reaction mixture was transferred to Sartorius Vivaspin-6 centrifuge tubes and centrifuged at 4500 rpm for 30 min at 22 °C to remove free PtMet2 and unreacted PAMAM-OH G2 (molecular weight 3272). The precipitate with a molecular weight of more than 5000 u remained on the membrane. The final product remaining on the membrane was dried in a vacuum at room temperature. The chemical shifts of the ^13^C NMR and ^1^H NMR spectra for the conjugate products were assigned by comparison with the spectrum PtMet2 and PAMAM-OH G2 dendrimer. In the ^13^C NMR spectrum, the appearance of a new ester carbonyl peak at 167.6 ppm for PtMet2–PAMAM conjugate indicated that PtMet2 was covalently bound to the PAMAM-OH G2 dendrimer.

Yield: 16.3% (6.7 mg); yellow powder; mp 246–249 °C; ^1^H-NMR (DMSO-d_6_) δ (ppm): 9.24 (br, s, amidine), 8.87–8.11 (m, CONH), 8.03 (s, 4H, Pz), 7.91 (d, *J* = 8.6 Hz, 8H, Ar), 7.68 (d, *J* = 8.3 Hz, 8H, Ar), 4.69 (s, 8H, CH_2_), 4.12 (m, CH_2_), 3.69 (m, 8H, CH_2_), 3.54–3.40 (m, NH-CH_2_-CH_2_), (2.56–2.50 (m, NH-CH2-CH2-NH), 2.46 (s, 12H, CH_3_); ^13^C NMR (DMSO-d_6_) δ (ppm): 175.6 (CONH), 167.6 (ester), 165.2 (amidine), 151.80 (Pz), 149.5 (Ar), 138.31 (Pz), 132.84 (Pz), 129.5 (Ar), 122.0 (Ar), 118.0 (Ar), 65.3 (CH_2_), 59.68 (CH_2_), 43.56 (CH_2_), 38.3 (CH_2_), 37.6 (CH_2_), 14.12 (CH_3_); IR (KBr, cm^−1^): 3359 (C=NH imine), 3143 (NH_3_^+^), 3208 (amide), 3131 (COOH), 2939 (CH_3_), 2857 (CH_2_), 1686 (C=O ester), 1647 (amide), 1607 (triazene), 1572 (CN imidazole ring), 1558 (amide), 1514 (NH_3_^+^), 1510 (N-O), 1485 (CH_2_/CH_3_), 1439 (CH_3_), 1380 (N-O), 1259 (ester), 1175 (triazene), 1164 (C-O ester), 854 (Ar-NO_2_), 525 (Pt-N); MS (ES, HR) m/z (M^+^) calcd. for C_350_H_464_N_146_O_92_Pt_8_ 9749.132, found 9746.52.

### 4.5. Cell Lines and Cell Culture

Human breast cancer cell lines (MCF-7 and MDA-MB-231) and normal human breast epithelial cells (MCF-10A) were purchased from the American Type Culture Collection (ATCC, Manassas, VA, USA). MCF-7 and MDA-MB-231 cells were cultured in Dulbecco’s modified eagle medium (Gibco, San Diego, CA, USA), MCF-10A cells were cultured in mammary epithelial cell growth medium with supplements: BPE, hEGF, insulin, hydrocortisone, GA-1000 (Lonza, Basel, Switzerland). All media were complemented by 10% of fetal bovine serum (FBS) and 1% of antibiotics: penicillin and streptomycin (both Gibco, San Diego, CA, USA). The cells were maintained in an incubator that provides the optimal growth conditions for the cell culture: 5% CO_2_, 37 °C, and humidity in a range of 90‒95%. The cells were cultured in 100 mm plates (Sarstedt, Newton, NC, USA). Subsequently after obtaining a subconfluent cell culture, the cells were detached with 0.05% trypsin with 0.02% EDTA (Gibco, San Diego, CA, USA). Then, utilizing a Scepter 3.0 handheld automated cell counter (Milipore, Burlington, MA, USA), the number of cells was quantified and seeded at a density of 5 × 10^5^ cells per well in six-well plates (“Nunc”) in 2 mL of the growth medium (Dulbecco’s modified eagle medium or mammary epithelial cell growth medium, respectively). In the present study, cells that obtained 80% of confluence were used.

### 4.6. Cell Viability Determination Using MTT Assay

The cytotoxic activity of novel synthesized compounds (PtMet2 and PtMet2–PAMAM) against human breast cancer cell lines (MCF-7 and MDA-MB-231) and normal human breast epithelial cells (MCF-10A) was analyzed by a spectrocolorimetric assay using thiazolyl blue tetrazolium bromide (MTT) in accordance with the procedure in the literature [9]. Briefly, MCF-7, MDA-MB-231, and MCF-10A cells were seeded in six-well plates “Nunc” at a density of 5 × 10^5^ cells/well and incubated for 24 h in optimal growth conditions (37 °C, 5% CO_2_). Subsequently, PtMet2, PtMet2–PAMAM, and the reference drug (cisplatin) at concentrations 0.1, 0.25, 0.5, 1.0, 1.5, 2.5, and 5.0 µM were added in duplicate, and the plates were incubated for another 24 h. Next, the plates were washed with PBS three times. Then, 1 mL PBS and 50 μL of 5 mg/cm^3^ MTT solution were added, and the incubation was continued for four hours. The MTT assay protocol is based on the enzymatic reduction of yellow soluble tetrazolium bromide to blue formazan, which occurs only in living cells. Then, the product is dissolved in DMSO, and the absorbance is measured at 570 nm wavelength. The absorbance result obtained in the control was taken as 100%, and the viability of the cells incubated with the tested compounds was shown as a percentage of the control cells. The analysis was performed using the Evolution 201 UV-visable spectrophotometer using the Thermo INSIGHT software (Thermo Fisher, Waltham, MA, USA, both).

### 4.7. Zebrafish Drug-Screening Assay

Zebrafish embryos were obtained from mating adults, maintained, and raised as described previously [36]. Zygote period cleaving eggs were transferred to six-well plastic cell culture plates filled with embryo medium E3. The eggs (10–12 per well) were exposed to PtMet2, PtMet2–PAMAM, and cisplatin (2.5 μM all of them) for three hours. The final volume of the medium in each well was 2 mL. DMSO was used as a drug solvent. The final concentration of DMSO in the wells did not exceed the damaging concentration of above 0.1%. The mock control embryos were incubated in embryo medium in the presence of 0.1% DMSO. The drug effect was recognized when all the eggs from one well changed in the same characteristic manner. Each experiment was carried out in three independent experiments. Observations of cell division and the development of the zebrafish eggs were done using a SteREO Discovery V8 stereo microscope (Zeiss, Jena, Germany) once every 15 min within the first three hours of incubation.

### 4.8. Flow Cytometry Assessment of Annexin V and Propidium Iodide Binding

Flow cytometry analysis for apoptosis induction by the tested compounds was quantified using the method described in the literature [55]. The test is based on the externalization of phosphatidylserine on the surface of apoptotic cells. In early and late apoptotic cells, the phosphatidylserine translocates to the outer layer of the membrane, which allows annexin V-FITC to attach. Propidium iodide is a fluorescent dye that penetrates into late apoptotic and necrotic cells with impaired cell membrane integrity. Hence, it is possible to distinguish four groups of cells: living cells, early apoptotic cells, late apoptotic cells, and necrotic cells. To calibrate the flow cytometer, two controls were performed: a positive control in which cells were treated with 3% formaldehyde solution to induce apoptosis and a negative control where cells had not been treated with any of the compounds.

MCF-7 and MDA-MB-231 cells were incubated (for 24 h) with PtMet2, PtMet2–PAMAM, and the reference drug (cisplatin). All compounds were used in two concentrations: 1.5 and 2.5 μM. Following incubation, the cells were dyed with FITC-labeled annexin V and propidium iodide. The analysis was performed on a BD FACSCanto II flow cytometer using the FACSDiva software (BD Biosciences Systems, San Jose, CA, USA, both). The equipment was calibrated with BD Cytometer Setup and Tracking Beads (BD Biosciences, San Diego, CA, USA).

### 4.9. Immunofluorescence Staining

The effect of the novel synthesized compounds on NF-κB, p38, PARP, POX, caspase-3, caspase-8, and caspase-9 protein expression was determined by immunofluorescence staining and cell bioimaging using the Pathway 855 fluorescence microscope, according to the method described in the literature [27]. Briefly, 100 μL of cells (4 × 10^4^ cells/well) were plated in a 96-well microplate and incubated in the appropriate conditions (37 °C, 5% CO_2_). Subsequently, after 24 h incubation during which cells reach about 80% confluence, the tested compounds were added, and incubation was continued for another 24 h. Afterwards, the medium was removed, and the cells were fixed with 100 μL of 3.7% formaldehyde solution in PBS for 10 min at room temperature. In the next step, the formaldehyde solution was removed, and the cells were rinsed three times with PBS buffer (3 × 100 µL). Then, 100 µL of 0.1% Triton X-100 was added to each well to perform permeabilization. The incubation was continued for five minutes at room temperature. Then, the cells were washed two times with PBS (100 µL/well). In the next step, non-specific protein binding sites were blocked by incubation with 3% FBS (100 µL/well) at room temperature for 30 min. The FBS solution was removed, and the cells were incubated with diluted (1:500) anti-NF-κB mouse monoclonal antibody, anti-p38 mouse monoclonal antibody, anti-PARP mouse monoclonal antibody, anti-POX mouse monoclonal antibody, anti-caspase-3 mouse monoclonal antibody, anti-caspase-8 mouse monoclonal antibody, and anti-caspase-9 mouse monoclonal antibody for one hour at room temperature. After incubation, the cells were washed three times with PBS buffer (3 × 100 µL) and then incubated in the dark with a FITC-labeled secondary anti-mouse antibody (1:1000) for one hour at room temperature. After incubation, the cells were washed again three times with PBS (3 × 100 µL) and incubated once more with Hoechst 33342 nuclear-staining solution in PBS (2 µg/mL). BD Pathway 855 confocal microscope (magnification × 400) was used to read the results using AttoVision software (BD Biosciences Systems, San Jose, CA, USA, both).

### 4.10. Intracellular Total ROS Activity Assay

To determine the effects of PtMet2 and PtMet2–PAMAM compounds on the oxidative stress of breast cancer cells MCF-7 and MDA-MB-231, a total ROS activity assay was performed according to the manufacturer’s protocol. The probe was a cell-permeant dye called Total ROS Green (Intracellular Total ROS Activity Assay; ImmunoChemistry Technologies, Bloomington, MN, USA). This dye quickly penetrates membrane structures and accumulates within the cell. In the presence of ROS, the non-fluorescent Total ROS Green dye molecule is oxidized by all various iterations of ROS molecular forms. In the oxidized state, the Total ROS Green dye molecule acquires fluorescence properties that enable its detection by flow cytometry as an indicator of the relative level of intracellular ROS activity. In short, after 24 h of incubation of breast cancer cells MCF-7 and MDA-MB-231 with the tested compounds at a concentration of 1.5 µM and 2.5  µM, the medium was removed, the cells were washed twice with cold PBS solution, and the assay buffer (provided by the kit manufacturer) was added. Then, the cells (at a density of 1 × 10^6^ cells/mL) were treated with 10 µL of Total ROS Green reagent and incubated for one hour at 37  °C in a CO_2_ incubator. After incubation, the cells were subjected to a rinsing procedure using the assay buffer. The cells were suspended in 500 µl of the assay buffer and analyzed using a flow cytometer (BD FACSCanto II flow cytometer) and FACSDiva software (both from BD Biosciences Systems, San Jose, CA, USA). The equipment was calibrated with the BD Cytometer Setup and Tracking Beads (BD Biosciences, San Diego, CA, USA).

### 4.11. Measuring the Number of Autophagosomes and Autolysosomes by Autophagy Assay

To determine the effects of PtMet2 and PtMet2–PAMAM compounds on the autophagy process of breast cancer cells MCF-7 and MDA-MB-231, an autophagy assay was performed according to the manufacturer’s protocol. The probe was a cell-permeant aliphatic molecule that fluoresces brightly when inserted in the lipid membranes of autophagosomes and autolysosomes (Autophagy Assay, Red kit; ImmunoChemistry Technologies, Bloomington, MN, USA). In short, the unfixed cells were washed and then resuspended in PBS with the added autophagy probe, Red solution. Afterwards, the cells were incubated for 30 min at 37 °C in the dark, washed, resuspended in cellular assay buffer, and the provided fixative was added at a volume/volume ratio of 1:5. The samples were measured immediately after preparation by flow cytometry using the BD FACSCanto II system (BD Biosciences Systems). The percentage of cells with autophagy was calculated using FACSDiva software (BD Biosciences Systems). The equipment was calibrated with the BD Cytometer Setup and Tracking Beads (BD Biosciences, San Diego, CA, USA).

### 4.12. Antibody Bax, LC3A/B and mTOR Detection

To check whether the tested compounds affect protein induction (Bax, LC3A/B, and mTOR in breast cancer cells MCF-7 and MDA-MB-231), the following were used: Bax antibody conjugated to FITC, LC3A/B antibody conjugated to PE, and mTOR antibody conjugated to Alexa Fluor 647, according to the manufacturer’s instructions. In brief, the centrifuged cells were resuspended in 4% formaldehyde and incubated for 15 min at room temperature. The cells were washed by centrifugation with excess PBS. Afterwards, permeabilization was performed by adding ice-cold 90% methanol to the cells and incubating for 60 min in an ice bath. The cells were washed by centrifugation with excess PBS again. Then, they were resuspended in 100 μL of diluted primary antibody, prepared in PBS at a 1:100 dilution and incubated for 60 min at room temperature in the dark, washed, and resuspended in 300 μL PBS. The samples were measured immediately after preparation by flow cytometry using BD FACSCanto II. An analysis of the results was performed using FACSDiva software (both from BD Biosciences Systems). The equipment was calibrated with the BD Cytometer Setup and Tracking Beads (BD Biosciences, San Diego, CA, USA).

### 4.13. Antibody AMPKβ1/2 and Beclin-1 Detection

For the purpose of identifying AMPKβ1/2 and Beclin-1 proteins in breast cancer cells MCF-7 and MDA-MB-231, AMPKβ1/2 and Beclin-1 antibodies were used according to the manufacturer’s instructions. In brief, the centrifuged cells were suspended in 4% formaldehyde and incubated for 15 min at room temperature. The cells were washed by centrifugation with excess PBS. Afterwards, permeabilization was performed by adding ice-cold 90% methanol to the cells and incubating for 60 min in an ice bath. The cells were washed by centrifugation with excess PBS again. Then, they were resuspended in 100 μL of diluted primary antibody, prepared in PBS at a 1:100 dilution, and incubated for 60 min at room temperature in the dark, washed and resuspended in 300 μL PBS. After that, they were suspended in 100 μL of diluted fluorochrome-conjugated secondary antibody (anti-mouse IgG Alexa Fluor^®^ 488 in the case of AMPKβ1/2 and anti-rabbit IgG Alexa Fluor 647 Conjugate in the case of Beclin-1) and incubated for 30 min at room temperature, protected from light. The samples were washed by centrifugation with excess PBS again and measured immediately by flow cytometry using BD FACSCanto II. An analysis of the results was performed using FACSDiva software (both from BD Biosciences Systems). The equipment was calibrated with the BD Cytometer Setup and Tracking Beads (BD Biosciences, San Diego, CA, USA).

### 4.14. Confocal Microscope Imaging

Imaging of MCF-7 and MDA-MB-231 cells was performed using the laser confocal microscope Nikon Eclipse Ti-E A1R-Si and Nikon NIS Elements AR software. The cells were cultured in sterile cover slips placed in Petri dishes to 80% confluence. Next, the cells were treated with the tested compounds. The examined cells were fixed with paraformaldehyde (4%). Cell membranes were permeabilized with 0.1% Triton X-100. All samples were blocked with glycine and BSA solutions and then incubated overnight at 4 °C with rabbit antibody conjugated with Phycoerythrin and Alexa Fluor 488: anti-p53 (1:50). Afterward, cells were stained with DAPI (1:500) and Phalloidin–Atto 565 (0.6 µM). The final staining allowed imaging the nucleus and actin filaments, respectively. Finally, the cover slips were mounted onto microscopic glass slides.

### 4.15. MDR Transporter Activity Assay

The MDR Assay kit (Abcam, Cambridge, UK) was used strictly following the manufacturer’s instructions. The test cells were washed twice with 5 mL of PBS by centrifugation at 1200 rpm for 10 min. Supernatants were discarded, and the cells were counted using a Scepter 3.0 handheld automated cell counter (Milipore, Burlington, MA, USA). Then, each of the tested samples was divided into three parts and analyzed. For measuring the activity of MDR1, MRP1, and BCRP (tubes 1–12), 5 μL of verapamil (MDR1 inhibitor) was added into tubes 1–3, 5 μL of MK-571 (MRP1 inhibitor) was added into tubes 4–6, 5 μL of noviobiocin (inhibitor BCRP) was added into tubes 7–9, and 125 μL DMEM with 1 μL DMSO was added into tubes 10–12. Then, the samples were incubated in 37 °C for five minutes. After this time, 125 μL of Efflux Green Detection Reagent was added into tubes 1–12, and the samples were incubated for 30 min at 37 °C. After 25 min of incubation, 5 μL propidium iodide was added to each tube. Thereafter, the samples were centrifuged for 10 min at 1200 rpm. Supernatants were discarded, and the cells were resuspended in 500 μL of PBS and run on the flow cytometer immediately. Measurements were conducted on a BD FACSCanto II flow cytometer using the FACSDiva software (BD Biosciences, San Diego, CA, USA, both). The equipment was calibrated with the BD Cytometer Setup and Tracking Beads (BD Biosciences, San Diego, CA, USA).

Multidrug resistance activity factor (MAF) was calculated from the difference between the mean fluorescence intensity (MFI) of cells with and without the highly selective inhibitors. Calculations followed these formulas:

**MAF_MDR1_** = 100 × (F_MDR1_ − F_0_)/F_MDR1_

**MAF_MRP_** = 100 × (F_MRP_ − F_0_)/F_MRP_

**MAF_BCRP_** = 100 × (F_BCRP_ − F_0_)/F_BCRP_

F_MDR_—MFI with MDR1 inhibitor (Verapamil)

F_MRP_—MFI with MRP inhibitor (MK-571)

F_BCRP_—MFI with BCRP inhibitor (Novobiocin)

F_0_—MFI without inhibitor

The theoretical range of the MAF values are between 0 and 100. Studies comparing MAF values with clinical response to a chemotherapeutic treatment suggest that a specimen with an MAF value of <20 can be regarded as multidrug resistance negative, while MAF values >25 are indicative of multidrug resistance positive specimens.

### 4.16. Statistical Analysis

All numerical data are presented as mean ± standard deviation (SD) from at least three independent experiments in duplicate. Statistical analysis was conducted using the GraphPad Prism 8 software (GraphPad Software, San Diego, CA, USA). Statistical differences in multiple groups were determined by one-way ANOVA followed by Tukey’s test *p* < 0.05 was considered statistically significant.

## 5. Conclusions

In this report, novel imidazole platinum(II) complexes were synthesized. Cytotoxicity evaluation identified the imidazole platinum(II) compound conjugated with dendrimer PAMAM (PtMet2–PAMAM) as the most promising compound with IC_50_ values lowered to 0.86 µM and 0.48 µM against MCF-7 and MDA-MB-231 cell lines, respectively. In the case of a free compound, IC_50_ for both breast cell lines was >5.0 µM. Furthermore, PtMet2–PAMAM also displayed promising antiproliferative properties in a zebrafish embryo model. In addition, the molecular mechanism studies showed that the tested compounds differ in their biological activity to apoptosis and autophagy. The study also revealed that PtMet2–PAMAM might exert its antitumor activity via activating the p53 and POX to cause an increase in the level of intracellular ROS, up-regulating the pro-apoptotic proteins and caspase cascade to mediate the intrinsic and extrinsic apoptosis. Moreover, the conjugate with the dendrimer significantly reduced the effect of drug efflux responsible for drug resistance. These results suggest that PtMet2–PAMAM may serve as a promising candidate in the development of a novel therapeutic agent to treat breast cancer.

## Figures and Tables

**Figure 1 ijms-22-05581-f001:**
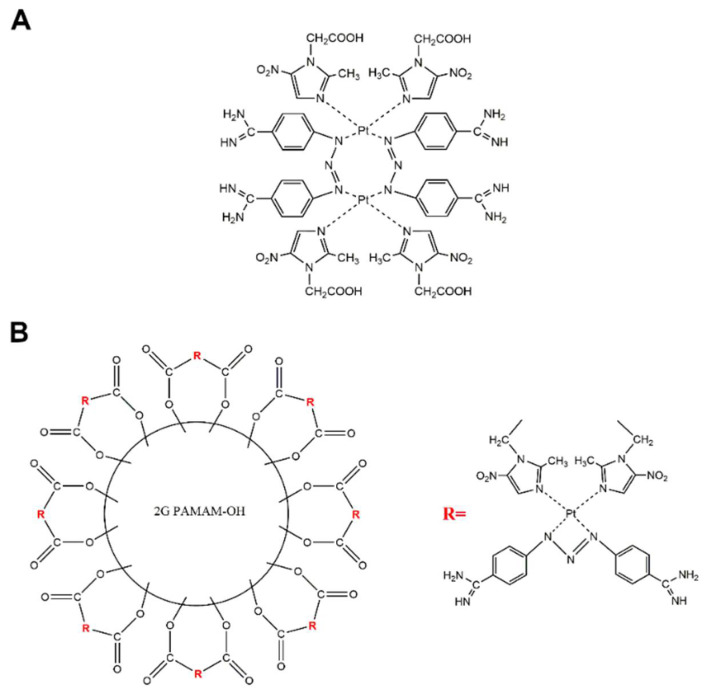
Chemical structures of PtMet2 (**A**) and PtMet2–PAMAM (**B**).

**Figure 2 ijms-22-05581-f002:**
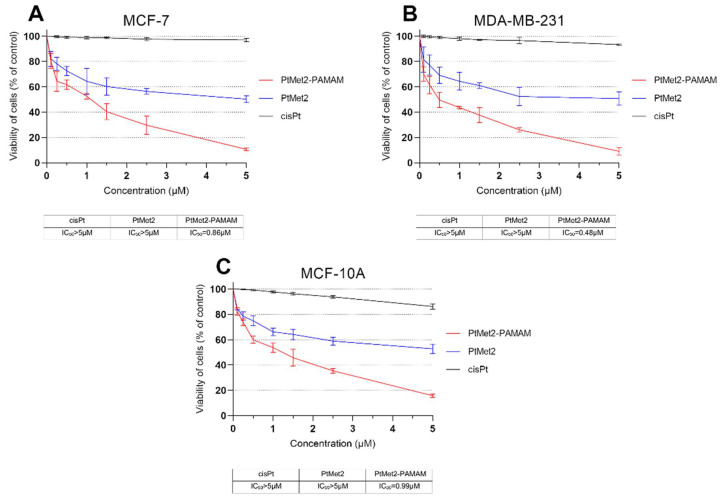
Viability of MCF-7 (**A**) and MDA-MB-231 (**B**) breast cancer cells and normal human breast epithelial cell MCF-10A (**C**) treated for 24 h with different concentrations of the tested compounds: PtMet2, PtMet2–PAMAM, and cisplatin. Mean values ± SD from three independent experiments (*n* = 3) done in duplicate are presented.

**Figure 3 ijms-22-05581-f003:**
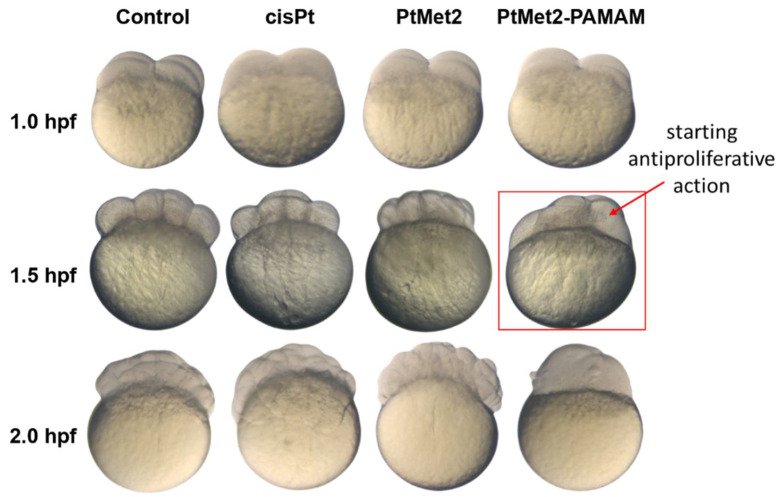
Effect of PtMet2, PtMet2–PAMAM, and cisplatin on cell division in the zebrafish embryo. Zebrafish eggs after 1.0, 1.5, and 2.0 h of exposure to the tested compounds; *n* = 10. hpf: hours post fertilization; hpt: hours post treatment.

**Figure 4 ijms-22-05581-f004:**
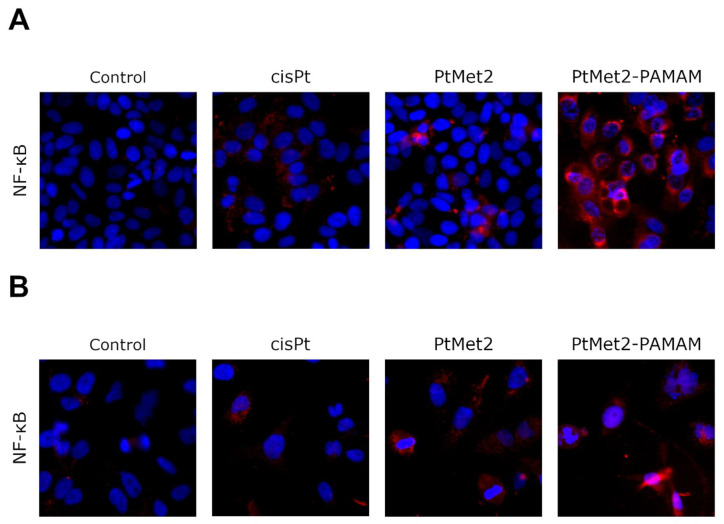
Expression and nuclear translocation of NF-κB detected by fluorescence microscopy in MCF-7 (**A**) and MDA-MB-231 (**B**) breast cancer cells after treatment with PtMet2, PtMet2–PAMAM, and cisplatin (2.5 μM all of them) for 24 h. Representative photographs are shown.

**Figure 5 ijms-22-05581-f005:**
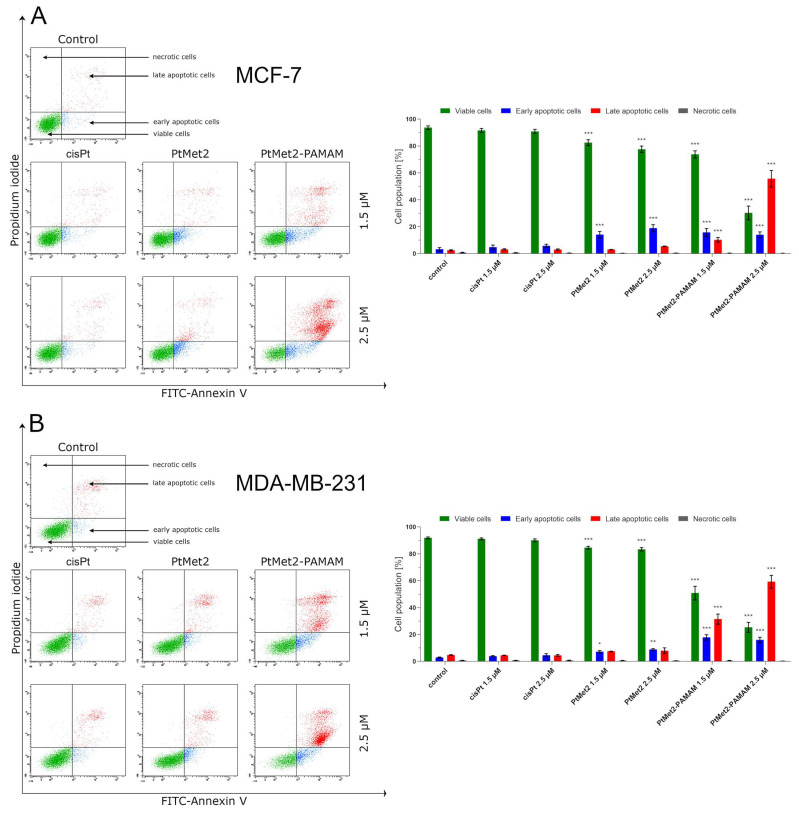
Flow cytometry analysis of MCF-7 (**A**) and MDA-MB-231 (**B**) breast cancer cells after 24 h incubation with PtMet2, PtMet2–PAMAM, and cisplatin (1.5 μM and 2.5 μM) and subsequent staining with Annexin V and propidium iodide. Mean percentage values from three independent experiments (*n* = 3) done in duplicate are presented. * *p* < 0.05 vs. control group, ** *p* < 0.01 vs. control group, *** *p* < 0.001 vs. control group.

**Figure 6 ijms-22-05581-f006:**
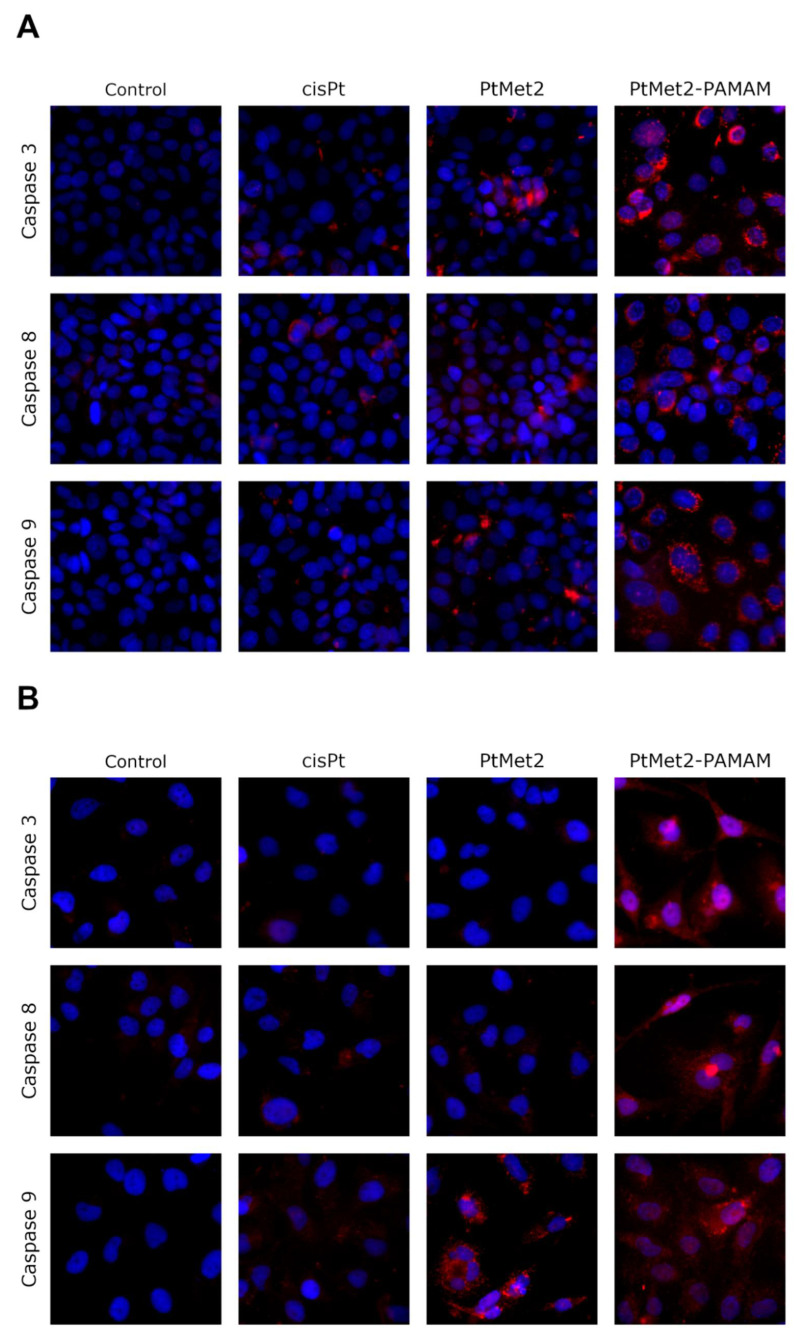
Expression and nuclear translocation of caspase-3, -8, and -9 detected by fluorescence microscopy in MCF-7 (**A**) and MDA-MB-231 (**B**) breast cancer cells after treatment with PtMet2, PtMet2–PAMAM, and cisplatin (2.5 μM all of them) for 24 h. Representative photographs are shown.

**Figure 7 ijms-22-05581-f007:**
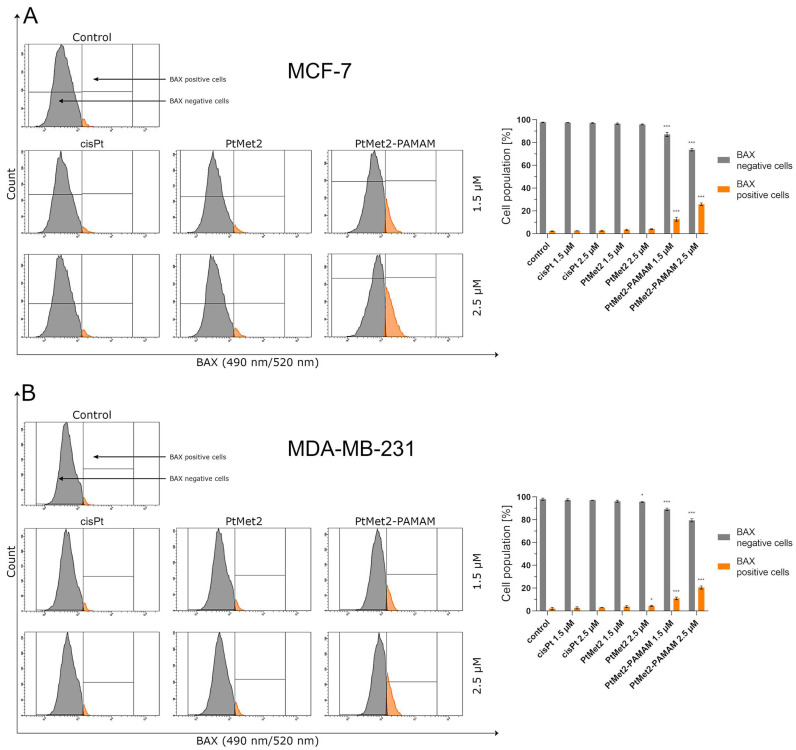
Anti-Bax antibody flow cytometric analysis of MCF-7 (**A**) and MDA-MB-231 (**B**) breast cancer cells (orange color) compared to a negative control cell (gray color) after 24 h of incubation with PtMet2, PtMet2–PAMAM, and cisplatin (1.5 and 2.5 μM). Mean percentage values from three independent experiments (*n* = 3) done in duplicate are presented. * *p* < 0.05 vs. control group, *** *p* < 0.001 vs. control group.

**Figure 8 ijms-22-05581-f008:**
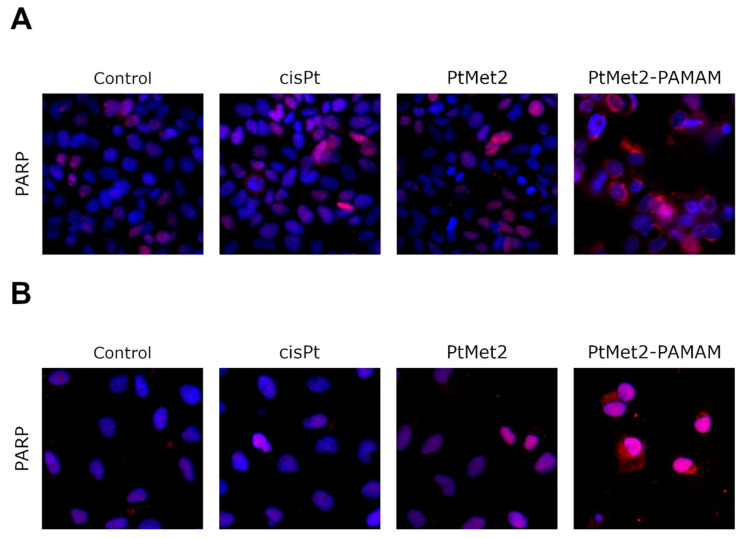
Expression and nuclear translocation of PARP detected by fluorescence microscopy in MCF-7 (**A**) and MDA-MB-231 (**B**) breast cancer cells after treatment with PtMet2, PtMet2–PAMAM, and cisplatin (2.5 μM all of them) for 24 h. Representative photographs are shown.

**Figure 9 ijms-22-05581-f009:**
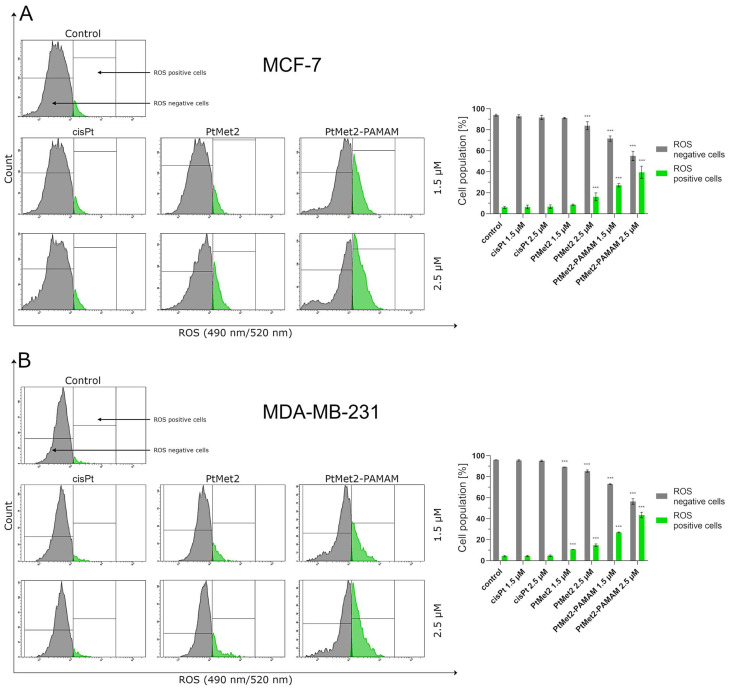
ROS induction in MCF-7 (**A**) and MDA-MB-231 (**B**) breast cancer cells measured by flow cytometry using Intracellular Total ROS Activity Assay (green color) compared to negative control cells (gray color) after 24 h incubation with PtMet2, PtMet2–PAMAM, and cisplatin (1.5 μM and 2.5 μM). Mean percentage values from three independent experiments (*n* = 3) done in duplicate are presented. *** *p* < 0.001 vs. control group.

**Figure 10 ijms-22-05581-f010:**
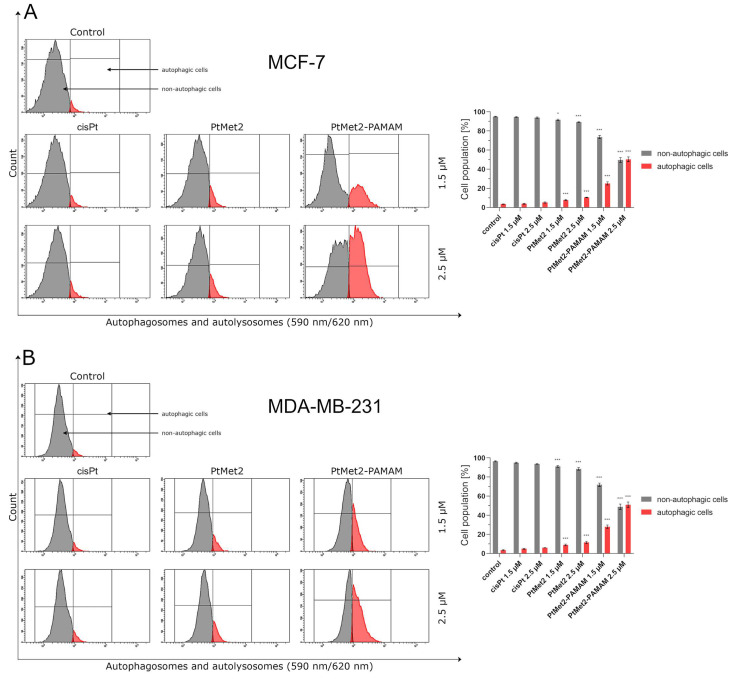
Autophagy induction in MCF-7 (**A**) and MDA-MB-231 (**B**) breast cancer cells measured by flow cytometry using Autophagy Probe (red color) compared to negative control cells (gray color) after 24 h of incubation with PtMet2, PtMet2–PAMAM, and cisplatin (1.5 μM and 2.5 μM). Mean percentage values from three independent experiments (*n* = 3) done in duplicate are presented. * *p* < 0.05 vs. control group, *** *p* < 0.001 vs. control group.

**Figure 11 ijms-22-05581-f011:**
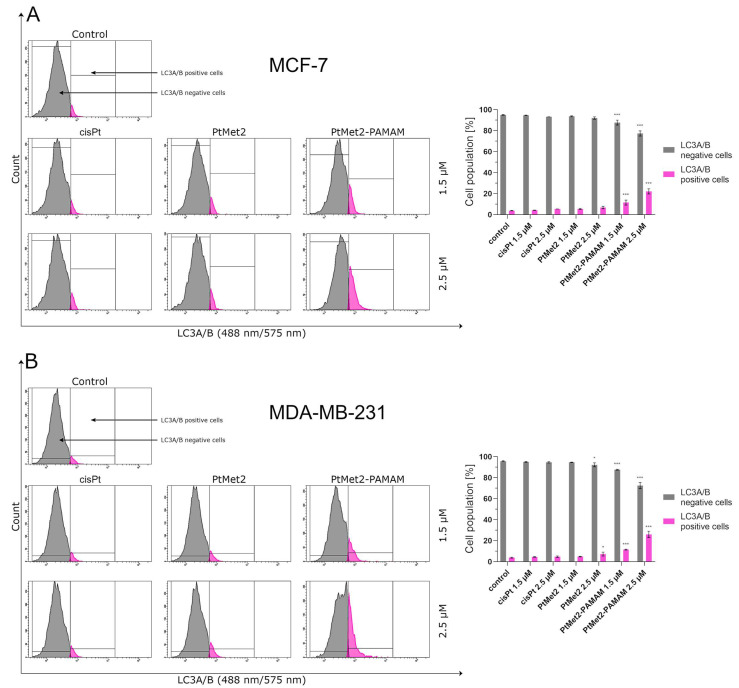
Anti-LC3A/B antibody flow cytometric analysis of MCF-7 (**A**) and MDA-MB-231 breast cancer cells (**B**) (pink color) compared to a negative control cell (gray color) after 24 h of incubation with PtMet2, PtMet2–PAMAM, and cisplatin (1.5 μM and 2.5 μM). Mean percentage values from three independent experiments (*n* = 3) done in duplicate are presented. * *p* < 0.05 vs. control group, *** *p* < 0.001 vs. control group.

**Figure 12 ijms-22-05581-f012:**
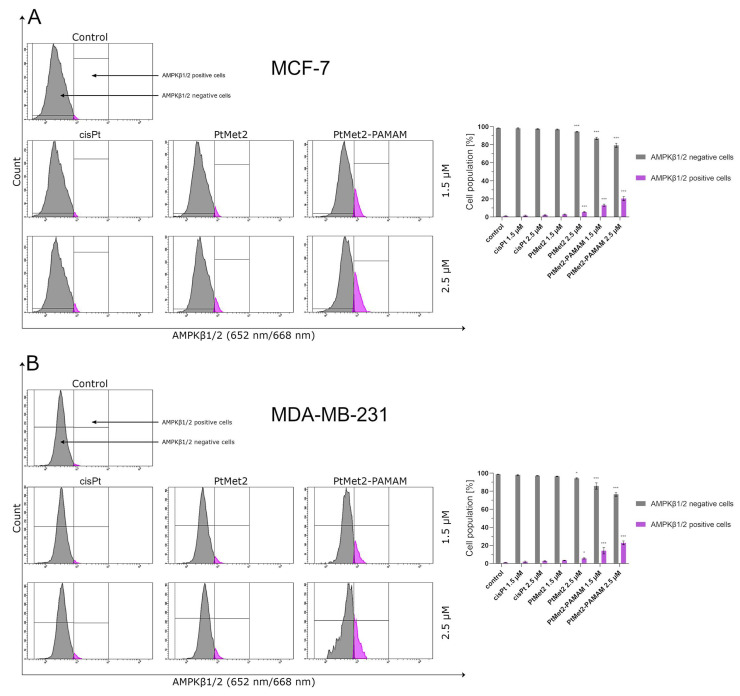
Anti-AMPKβ1/2 antibody flow cytometric analysis of MCF-7 (**A**) and MDA-MB-231 breast cancer cells (**B**) (right histogram—purple color) compared to a negative control cell (left histogram—gray color) after 24 h of incubation with PtMet2, PtMet2–PAMAM, and cisplatin (1.5 μM and 2.5 μM). Mean percentage values from three independent experiments (*n* = 3) done in duplicate are presented. * *p* < 0.05 vs. control group, *** *p* < 0.001 vs. control group.

**Figure 13 ijms-22-05581-f013:**
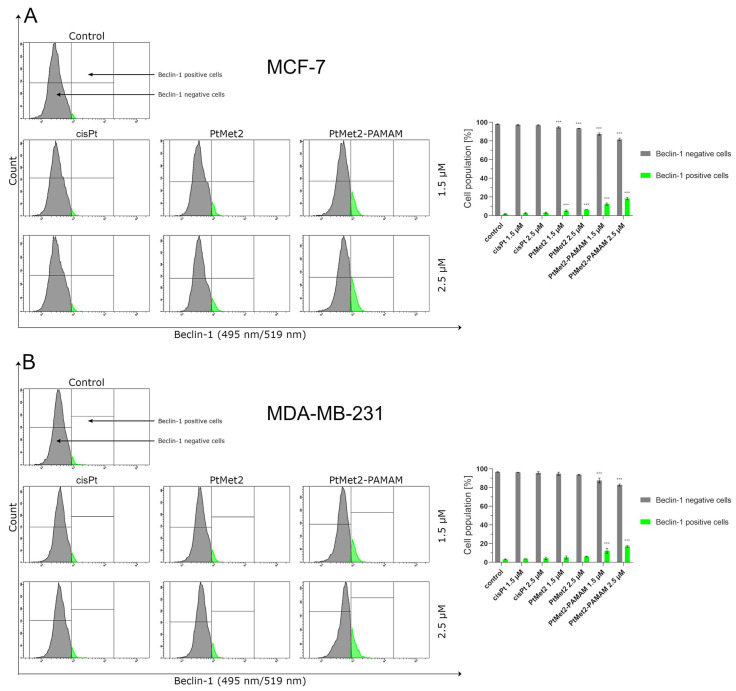
Anti-Beclin-1 antibody flow cytometric analysis of MCF-7 (**A**) and MDA-MB-231 breast cancer cells (**B**) (light green color) compared to a negative control cell (gray color) after 24 h of incubation with PtMet2, PtMet2–PAMAM, and cisplatin (1.5 μM and 2.5 μM). Mean percentage values from three independent experiments (*n* = 3) done in duplicate are presented. *** *p* < 0.001 vs. control group.

**Figure 14 ijms-22-05581-f014:**
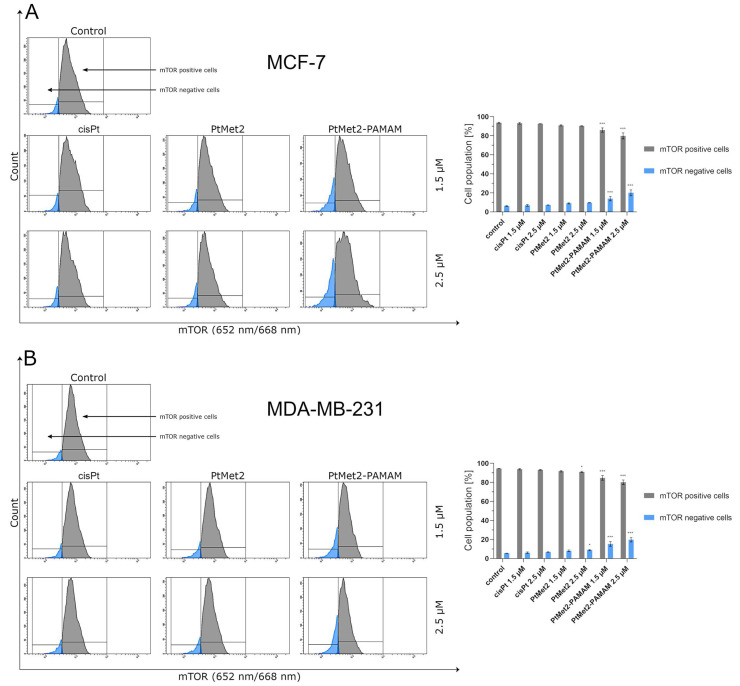
Anti-mTOR antibody flow cytometric analysis of MCF-7 (**A**) and MDA-MB-231 breast cancer cells (**B**) (blue color) compared to a negative control cell (gray color) after 24 h of incubation with PtMet2, PtMet2–PAMAM, and cisplatin (1.5 μM and 2.5 μM). Mean percentage values from three independent experiments (*n* = 3) done in duplicate are presented. * *p* < 0.05 vs. control group, *** *p* < 0.001 vs. control group.

**Figure 15 ijms-22-05581-f015:**
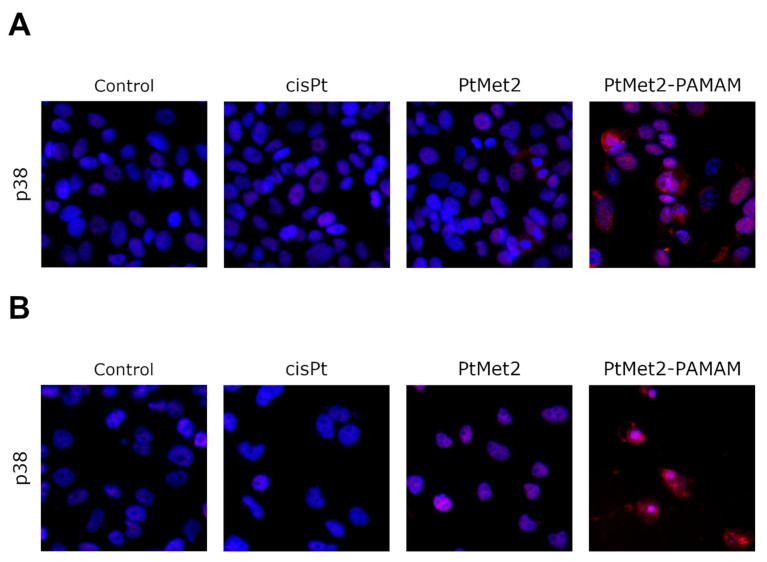
Expression and nuclear translocation of p38 detected by fluorescence microscopy in MCF-7 (**A**) and MDA-MB-231 (**B**) breast cancer cells after treatment with PtMet2, PtMet2–PAMAM, and cisplatin (2.5 μM all of them) for 24 h. Representative photographs are shown.

**Figure 16 ijms-22-05581-f016:**
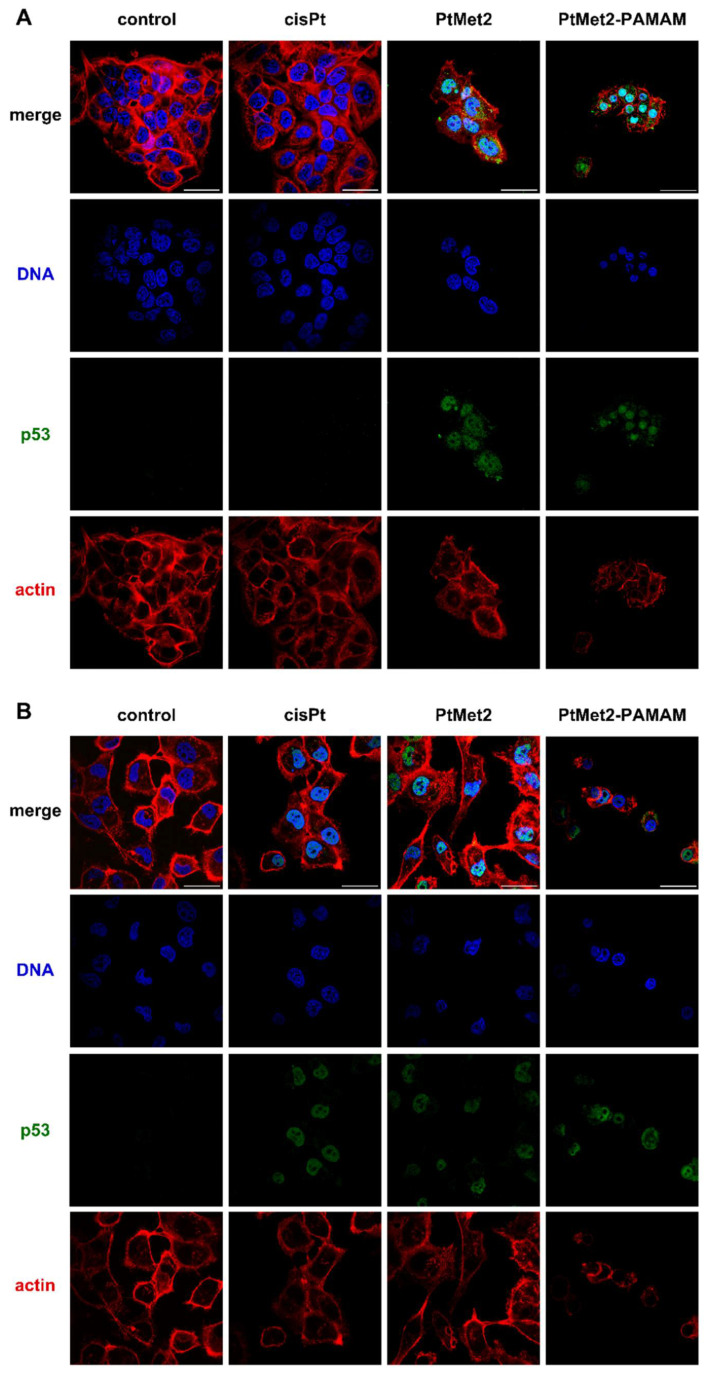
Confocal micrographs of MCF-7 (**A**) and MDA-MB-231 (**B**) breast cancer cells treated with PtMet2, PtMet2–PAMAM, and cisplatin (2.5 μM all of them, incubation time 24 h). For each panel, the images from bottom to top show actin filaments, p53, DNA, and overlays of the four images, scale bar 50 µm.

**Figure 17 ijms-22-05581-f017:**
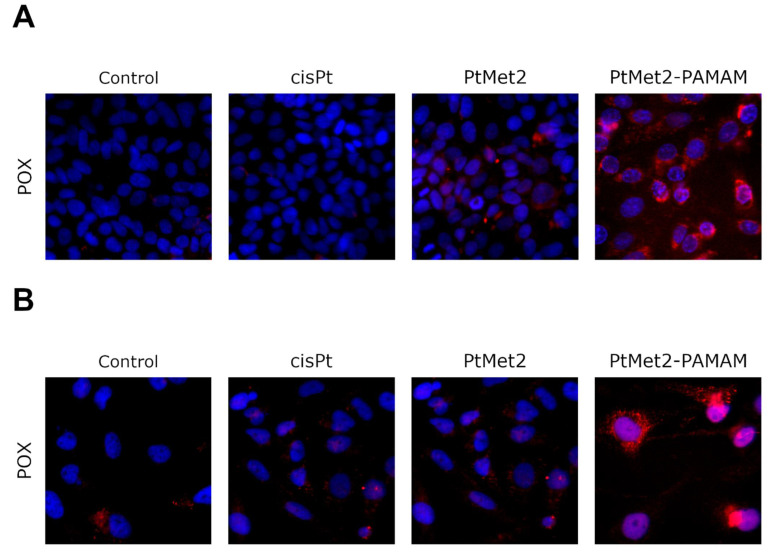
Expression and nuclear translocation of POX detected by fluorescence microscopy in MCF-7 (**A**) and MDA-MB-231 (**B**) breast cancer cells after treatment with PtMet2, PtMet2–PAMAM, and cisplatin (2.5 μM all of them) for 24 h. Representative photographs are shown.

**Figure 18 ijms-22-05581-f018:**
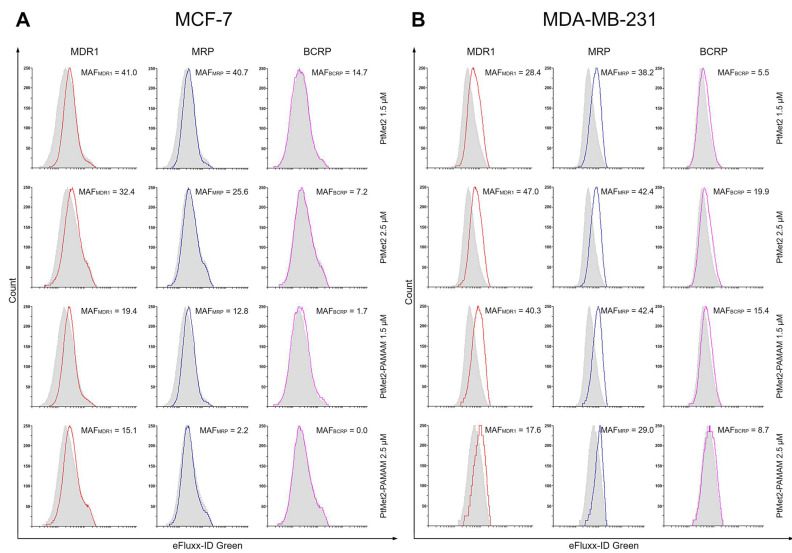
Flow cytometry analysis of MCF-7 (**A**) and MDA-MB231 (**B**) cells treated for 24 h with PtMet2 and PtMet2–PAMAM (1.5 μM and 2.5 μM, both). Cells incubated with Efflux Green Detection Reagent with and without specific inhibitors (Verapamil, MK-571 and Novoiobin). Gray histograms show the fluorescence of the inhibitor-treated samples and tinted histograms show the fluorescence of the treated cells (MDR1—red histogram, MRP—blue histogram, BCRP—purple histogram). The difference in fluorescence is indicative of corresponding protein activity. The numbers in the upper right corners are MAF scores (multidrug resistance activity factors), quantitative characteristics of multidrug resistance.

**Figure 19 ijms-22-05581-f019:**
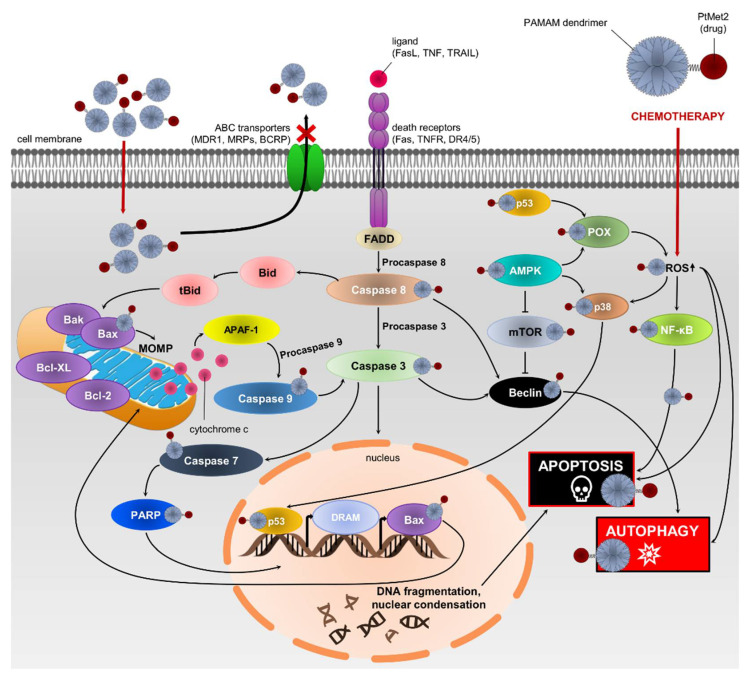
Schematic representation of possible anticancer mechanism action of PtMet2–PAMAM in MCF-7 and MDA-MB-231 cells.

## Data Availability

Department of Synthesis and Technology of Drugs, Medical University of Bialystok, Kilinskiego 1, 15-089 Bialystok, Poland.

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
