# Peer review of "Mechanism of Anticancer Action of Novel Imidazole Platinum(II) Complex Conjugated with G2 PAMAM-OH Dendrimer in Breast Cancer Cells"

_ijms, 2021, doi:10.3390/ijms22115581_

Round 1

Reviewer 1 Report

the manuscript “Mechanism of Anticancer Action of Novel Imidazole Platinum (II) Complex Conjugated with G2 PAMAM-OH Dendrimer in Breast Cancer Cells” is interesting and well conducted on cells express both the wild type and variant oestrogen receptors as well as progesterone receptor and on a highly aggressive, invasive and poorly differentiated triple-negative breast cancer cell line.

It lacks a real clinical relapse (in which molecular subtype of BC, in which setting is it potentially indicated the Platinum (II) Complex Conjugated?)

In this perspective, I would like to suggest to evaluate whether their model on cell lines is capable of inducing as well as a biological activity to apoptosis and autophagy an immunogenic cell death. This would provide the rationale not only for efficacy and tolerability but above all a combination role (as with nab-paclitaxel) for combination treatments with immunotherapy in triple negative tumors ... (Schmid N Engl J Med 2018);

  minor revision

Line 38:  specify the gender

Line 40:  specify that the mortality rate of breast cancer actually show a decrease linked both to an improvement in early diagnosis and to a precision treatment in the different molecular subtypes -> hence the need to introduce new and more active therapeutic strategies to improve the impact on OS

Line 43: The standard drug used for the systemic treatment of BC is cisplatin (cisPt)”

this sentence is not completely true and needs to be corrected. Indeed, platinum salts are mainly used to treat the triple negative breast cancer, but only a subgroup of triple negative patients (carriers of BRCA1 / 2 gene mutations or repair deficiency or BRCAnes) obtain a concrete benefit from these drugs (TNT study). Therefore, cisplatin is not the standard drug used in all BC subtypes.

(I would like to advise you to include a clinical oncologist in your very interesting research group).  

Author Response

Dear Editor,

Please find attached response to Reviewer 1.

Sincerely,

Robert Czarnomysy

Reviewer 2 Report

The authors synthetize a novel class of dendrimers carrying Pt(II) as a potential antitumour agent. The work is of intentest, deep and overall well-conducted.

They tested their activity against two breast cancer lines and using as controls breast epithelial cells. Although the PtMet2-PAMAM works better that Cisplatin or PtMet2, there is no clear difference in the antriproliferative activity of PtMet2-PAMAM between cancer and control cells (Figure 2). They confirm this antiproliferative activity in zebrafish embryos (Figure 3). They then carry out a quite detailed characterization of the action mechanism of PtMet2-PAMAM, including the activation of NF-kB, apoptosis, ROS production………

However, my only major concern is the selectivity. In Figure 2, the efficacy of PtMet2-PAMAM is tested against cancer and control cells, and the results are very similar. Then, the rest of the work (except the zebrafish) is done with cancer cells. Therefore, PtMet2-PAMAM is better in delivery but not in selectivity (potentially kills the same cancer and non-cancer cells). The authors should discuss this in detail.

Author Response

Dear Editor,

Please find attached response to Reviewer 2.

Sincerely,

Robert Czarnomysy
